# Deep Policy Gradient Methods Without Batch Updates, Target Networks, or Replay Buffers

**Gautham Vasan**[12]    **Mohamed Elsayed**[12]    **Alireza Azimi**[*12]    **Jiamin He**[*12]

**Fahim Shariar**[12]    **Colin Bellinger**[3]    **Martha White**[124]    **A. Rupam Mahmood**[124]

[1]University of Alberta    [2]Amii    [3]National Research Council of Canada    [4]CIFAR Canada AI Chair
{vasan, mohamedelsayed, sazimi, jiamin12, fshahri1}@ualberta.ca
colin.bellinger@nrc-cnrc.gc.ca   {whitem, armahmood}@ualberta.ca

## Abstract

Modern deep policy gradient methods achieve effective performance on simulated robotic tasks, but they all require large replay buffers or expensive batch updates, or both, making them incompatible for real systems with resource-limited computers. We show that these methods fail catastrophically when limited to small replay buffers or during *incremental learning*, where updates only use the most recent sample without batch updates or a replay buffer. We propose a novel incremental deep policy gradient method — *Action Value Gradient (AVG)* and a set of normalization and scaling techniques to address the challenges of instability in incremental learning. On robotic simulation benchmarks, we show that AVG is the only incremental method that learns effectively, often achieving final performance comparable to batch policy gradient methods. This advancement enabled us to show for the first time effective deep reinforcement learning with real robots using only incremental updates, employing a robotic manipulator and a mobile robot.[1]

## 1   Introduction

Real-time or online learning is essential for intelligent agents to adapt to unforeseen changes in dynamic environments. However, real-time learning faces substantial challenges in many real-world systems, such as robots, due to limited onboard computational resources and storage capacity (Hayes and Kanan 2022, Wang et al. 2023, Michieli and Ozay 2023). The system must process observations, compute and execute actions, and learn from experience, all while adhering to strict computational and time constraints (Yuan and Mahmood 2022). For example, the Mars rover faces stringent limitations on its computational capabilities and storage capacity (Verma et al. 2023), constraining the system's ability to run computationally intensive algorithms onboard.

Deep policy gradient methods have risen to prominence for their effectiveness in real-world control tasks, such as dexterous manipulation of a Rubik's cube (Akkaya et al. 2019), quadruped dribbling of a soccer ball (Ji et al. 2023), and magnetic control of tokamak plasmas (Degrave et al. 2022). These methods are typically used offline, such as in simulations, as they have steep resource requirements due to their use of large storage of past experience in a replay buffer, target networks and computationally intensive batch updates for learning. As a result, these methods are ill-suited for on-device learning and generally challenging to use for real-time learning. To make these methods applicable to resource-limited computers such as edge devices, a natural approach is to reduce the replay buffer size, eliminate target networks, and use smaller batch updates that meet the resource constraints.

In Figure 1, we demonstrate using four MuJoCo tasks (Todorov et al. 2012) that the learning performance of batch policy gradient methods degrades substantially when the replay buffer size is

---

[1]Code: `https://github.com/gauthamvasan/avg`    *Equal Contributions.

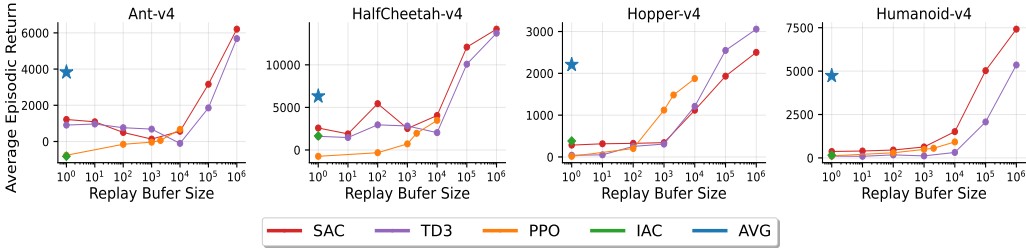

Figure 1: Impact of reducing replay buffer size on SAC, PPO, and TD3: Decreasing the replay buffer size adversely affects learning. In contrast, AVG succeeds despite learning without a replay buffer, as shown by a "buffer size" of 1 in the plots. Each data point represents the mean episodic return over the final 100K steps, averaged across 30 runs. All methods were trained for 10M timesteps.

reduced from their large default values. Specifically, Proximal Policy Optimization (PPO, Schulman et al., 2017), Soft Actor-Critic (SAC, Haarnoja et al., 2018), and Twin Delayed Deep Deterministic Policy Gradient (TD3, Fujimoto et al., 2018) fail catastrophically when their buffer size is reduced to 1. This case corresponds to *incremental learning*, where learning relies solely on the most recent sample, thus precluding the use of a replay buffer or batch updates.

Incremental learning methods (Vijayakumar et al. 2005, Mahmood 2017) are computationally cheap and commonly used for real-time learning with linear function approximation (Degris et al. 2012, Modayil et al. 2014, Vasan and Pilarski 2017). However, incremental policy gradient methods, such as the incremental one-step actor-critic (IAC, Sutton and Barto 2018), are rarely used in applications of deep reinforcement learning (RL), except for a few works (e.g., Young and Tian 2019) that work in limited settings. The results in Fig. 1 indicate that their absence is due to their difficulty in learning effectively when used with deep neural networks. A robust incremental method that can leverage deep neural networks for learning in real-time remains an important open challenge.

Incremental policy gradient methods, such as IAC, employ the likelihood ratio gradient (LG) estimator to estimate the gradient. An alternative approach to estimating the gradient, the reparameterization gradient (RG) estimator or the pathwise gradient estimator, has been observed to demonstrate lower variance in practice and can effectively handle continuous state and action spaces (Greensmith et al. 2004, Fan et al. 2015, Lan et al. 2022). RG estimators have recently gained interest in RL due to their use in deep policy gradient methods such as TD3 and SAC. However, we currently lack incremental policy gradient methods that use the RG estimator.

We present a novel incremental algorithm, called *Action Value Gradient (AVG)*, which leverages deep neural networks and utilizes the RG estimator. While batch updates, replay buffers, and target networks are required to stabilize deep RL (D'Oro et al. 2022, Schwarzer et al. 2023), AVG instead incorporates normalization and scaling techniques to learn stably in the incremental setting (see Sec. 3). In Sec. 4, we demonstrate that AVG achieves strong results across a wide range of benchmarks, being the only incremental algorithm to avoid catastrophic failure and learn effectively. In Sec. 5, we highlight the key challenges of incremental learning stemming from the large and noisy gradients inherent to the process. Through an ablation study, we discuss how normalization and scaling techniques help mitigate these issues for AVG and how they may salvage the performance of other methods, including IAC and an incremental variant of SAC. We also show that target networks hinder the learning performance of AVG in the incremental setting, with only aggressive updates of the target network towards the critic achieving results comparable to AVG, while their removal reduces memory demands and simplifies our algorithm. Finally, we apply AVG to real-time robot learning tasks, showcasing the first successful demonstration of an incremental deep RL method on real robots.

## 2  Background

We consider the reinforcement learning setting where an agent-environment interaction is modeled as a continuous state and action space Markov Decision Process (MDP) (Sutton and Barto 2018). The state, action, and reward at timestep $t \in (0, 1, 2, \dots)$ is denoted by $S_t \in \mathcal{S}$, $A_t \in \mathcal{A}$ and $R_{t+1} \in \mathbb{R}$ respectively. We focus on the episodic setting where the goal of the agent is to maximize the discounted return $G_t = \sum_{k=0}^{T-t-1} \gamma^k R_{t+k+1}$, where $\gamma \in [0, 1]$ is a discount factor and $T$ is the episode horizon. The agent selects an action $A_t$ according to a policy $\pi(\cdot|S_t)$ where $\pi(A|S)$

gives the probability of sampling an action $A$ in state $S$. Value functions are defined to be expected total discounted rewards from timestep $t$: $v_\pi(s) = \mathbb{E}_\pi \left[ \sum_{k=0}^{T-t-1} \gamma^k R_{t+k+1} | S_t = s \right]$ and $q_\pi(s, a) = \mathbb{E}_\pi \left[ \sum_{k=0}^{T-t-1} \gamma^k R_{t+k+1} | S_t = s, A_t = a \right]$. Our goal is to find the weights $\theta$ of a parameterized policy $\pi_\theta$ such that it maximizes the expected return starting from initial states: $J(\theta) \doteq \mathbb{E}_{S \sim d_0}[v_{\pi_\theta}(S)]$.

Parameterized policies are typically learned based on the gradients of $J(\theta)$. Since the true gradients $\nabla_\theta J(\theta)$ are typically not available, sample-based methods are commonly used for gradient estimation (Greensmith et al. 2004). Two existing theorems, known as *policy gradient theorem* and *reparameterization gradient theorem* provide ways of computing unbiased estimates of the gradient based on likelihood gradient (LG) estimators and reparameterization gradient (RG) estimators, respectively.

LG estimators use the log-derivative technique to provide an unbiased gradient estimate (Glynn 1990, Williams and Peng 1991): $\nabla_\theta \mathbb{E}_{p_\theta}[\phi(X)] = \mathbb{E}_{X \sim p_\theta}[\phi(X) \nabla_\theta \log p_\theta(X)]$, where $p_\theta(x)$ is the probability density of $x$ with parameters $\theta$, and $\phi(x)$ is a scalar-valued function. In the context of the policy gradient theorem (Sutton et al. 1999), the LG estimator is utilized to adjust the parameters $\theta$ of a policy $\pi$, in expectation, in the direction of the gradient of the expected return: $\nabla_\theta J(\theta) \propto \mathbb{E}_{S \sim d_{\pi,\gamma}, A \sim \pi_\theta}[\nabla_\theta \log \pi_\theta(A|S) q_{\pi_\theta}(S, A)]$, where $d_{\pi,\gamma}$ is the discounted stationary state distribution (Che et al. 2023). Many algorithms, including incremental ones like one-step actor-critic (IAC) and batch methods like A2C (Mnih et al. 2016), ACER (Wang et al. 2016) and PPO, are based on the policy gradient theorem and use the LG estimator.

RG estimators, also known as pathwise gradient estimators (Greensmith et al. 2004, Parmas and Sugiyama 2021), leverage the knowledge of the underlying density $p_\theta(x)$ by introducing a simpler, equivalent sampling procedure: $X \sim p_\theta(\cdot) = f_\theta(\xi), \xi \sim g(\cdot)$, where $\xi$ is sampled from a base distribution $g(\xi)$ independent of $\theta$, and $f_\theta$ is a function that maps $\xi$ to $X$. RG estimation can be written as $\nabla_\theta \mathbb{E}_{p_\theta}[\phi(X)] = \mathbb{E}_{\xi \sim g}[\nabla_\theta \phi(f_\theta(\xi))]$. RG estimators form the foundation of several batch RL algorithms, including Reward Policy Gradient (Lan et al. 2022), SAC and TD3. Lan et al. (2022) showed how RG estimation can be used to provide an alternative approach to unbiased estimation of the policy gradient through the reparametrization gradient theorem: $\nabla_\theta J(\theta) = \mathbb{E}_{S \sim d_{\pi,\gamma}, A \sim \pi_\theta} \left[ \nabla_\theta f_\theta(\xi; S)|_{\xi = h_\theta(A;S)} \nabla_A q_{\pi_\theta}(S, A) \right]$, where $h$ is a inverse function of $f$.

Deep reinforcement learning (RL) methods that use LG or RG estimators can often converge prematurely to sub-optimal policies (Mnih et al. 2016) or settle on a single output choice when multiple options could maximize the expected return (Williams and Peng 1991). This issue can be mitigated through *entropy regularization*, which promotes exploration and smoothens the optimization landscape under certain scenarios (Ahmed et al. 2019). This is accomplished by augmenting the reward function with an entropy term (i.e., $\mathbb{E}[-\log p_\theta(X)]$), encouraging the policy to maintain randomness in action selection. In this approach, the value functions are redefined as follows (Ziebart et al. 2010): $v_\pi^{\text{Ent}}(s) = \mathbb{E}_\pi \left[ \sum_{k=0}^{T-t-1} \gamma^k \left( R_{t+k+1} + \eta \mathcal{H}(\pi(\cdot|S_{t+k})) \right) | S_t = s \right]$, and $q_\pi^{\text{Ent}}(s, a) = \mathbb{E}_\pi \left[ R_{t+1} + \gamma v_\pi^{\text{Ent}}(S_{t+1}) | S_t = s, A_t = a \right]$, where $\eta$ is the entropy coefficient and entropy $\mathcal{H}(\pi(\cdot|s)) = -\int_\mathcal{A} \pi(a|s) \log \pi(a|s) da$.

## 3 The Action Value Gradient Method

In this section, we introduce a novel algorithm called Action Value Gradient (AVG, see Alg. 1)[2], outlining its key components and functionality and briefly discussing its theoretical foundations. We also discuss additional design choices that are crucial for robust and effective policy learning. AVG uses RG estimation, extended to incorporate entropy-augmented value functions:

$$\nabla_\theta J(\theta) = \mathbb{E}_{S \sim d_{\pi,\gamma}, A \sim \pi_\theta} \left[ \nabla_\theta f_\theta(\xi; S)|_{\xi = h_\theta(A;S)} \nabla_A \left( q_{\pi_\theta}(S, A) - \eta \log \left( \pi_\theta(A|S) \right) \right) \right]. \quad (1)$$

A brief derivation of this statement is provided in Appendix A.

The AVG algorithm maintains a parameterized policy or actor $\pi_\theta(A|S)$ to sample actions from a continuous distribution and critic $Q_\phi(S, A)$ that estimates the entropy-augmented action-value function. Both networks are parameterized using deep neural networks. AVG samples actions using the reparameterization technique (Kingma and Welling 2013), which allows the gradient to flow

---

[2]We also share a quick and easy-to-use implementation in the form of a python notebook on Google Colab

through the sampled action $A_\theta$ to the critic $Q_\phi(S, A_\theta)$, enabling the policy parameters $\theta$ to be updated smoothly based on the critic.

We use the same action $A_\theta$ to update both the actor and critic networks. First, the critic weights $\phi$ are updated using the temporal difference error; $\alpha_Q > 0$ is its step size. This step also involves sampling another action $A'$ that is used to estimate the bootstrap target. Then, the actor updates its weights $\theta$ based on $Q_\phi(S, A_\theta)$ and the sample entropy $-\log(\pi_\theta(A_\theta|S))$; $\alpha_\pi > 0$ is the step size of the actor, and $\eta \geq 0$ is used to weight the sample entropy term.

A careful reader may notice the similarity between the learning updates of SAC and AVG. However, SAC is an off-policy batch method, while AVG is an incremental on-policy method. SAC samples actions and stores them in a replay buffer. Unlike AVG, SAC does not reuse the same action to back-propagate gradients for the actor. Additionally, AVG is simpler than SAC, as it avoids the use of double Q-learning or target Q-networks (Van Hasselt et al. 2016) for stability. For comparison, we provide the pseudocode of an incremental variant of SAC, termed *SAC-1* (Alg. 5).

We also use *orthogonal initialization* (Saxe et al. 2013), *entropy regularization*, a *squashed normal policy*, as is standard in off-policy actor-critic methods like DDPG, TD3, and SAC. To enforce action bounds,

---

**Algorithm 1** Action Value Gradient (AVG)

**Initialize** $\gamma, \eta, \alpha_\pi, \alpha_Q$
$\theta, \phi$ with penultimate normalization
$n \leftarrow 0, \mu \leftarrow 0, \overline{\mu} \leftarrow 0$
$\boldsymbol{n}_\delta \leftarrow [0,0,0], \boldsymbol{\mu}_\delta \leftarrow [0,0,0], \overline{\boldsymbol{\mu}}_\delta \leftarrow [0,0,0]$
**for** however many episodes **do**
    Initialize S (first state of the episode)
    $S, n, \mu, \overline{\mu}, \_ \leftarrow \texttt{Normalize}(S, n, \mu, \overline{\mu})$
    $G \leftarrow 0$
    **while** S is not terminal **do**
        $A_\theta = f_\theta(\epsilon; S)$ where $\epsilon \sim \mathcal{N}(0, 1)$
        Take action $A_\theta$, observe $S', R$
        $S', n, \mu, \overline{\mu}, \_ \leftarrow \texttt{Normalize}(S', n, \mu, \overline{\mu})$
        $\sigma_\delta, \boldsymbol{n}_\delta, \boldsymbol{\mu}_\delta, \overline{\boldsymbol{\mu}}_\delta \leftarrow$
            $\texttt{ScaleTDError}(R, \gamma, \emptyset, \boldsymbol{n}_\delta, \boldsymbol{\mu}_\delta, \overline{\boldsymbol{\mu}}_\delta)$
        $G \leftarrow G + R$
        $A' \sim \pi_\theta(\cdot|S')$
        $\delta \leftarrow R + \gamma(Q_\phi(S', A') - \eta \log \pi_\theta(A'|S'))$
            $-Q_\phi(S, A_\theta)$
        $\delta \leftarrow \delta / \sigma_\delta$
        $\phi \leftarrow \phi - \alpha_Q \delta \nabla_\phi Q_\phi(S, a)|_{a=A_\theta}$
        $\theta \leftarrow \theta + \alpha_\pi \nabla_\theta(Q_\phi(S, A_\theta) - \eta \log \pi_\theta(A_\theta|S))$
        $S \leftarrow S'$
    **end while**
    $\sigma_\delta, \boldsymbol{n}_\delta, \boldsymbol{\mu}_\delta, \overline{\boldsymbol{\mu}}_\delta \leftarrow$
        $\texttt{ScaleTDError}(R, 0, G, \boldsymbol{n}_\delta, \boldsymbol{\mu}_\delta, \overline{\boldsymbol{\mu}}_\delta)$
**end for**

---

a squashed normal policy passes the sampled action from a normal distribution through the $\texttt{tanh}$ function to obtain actions in the range $[-1, 1]$: $A_\theta = f_\theta(\xi; S) = \texttt{tanh}(\mu_\theta(S) + \xi\sigma_\theta(S))$ where $\xi \sim \mathcal{N}(0, 1)$. This parameterization is particularly useful for entropy-regularized RL objectives. In an unbounded normal policy, the standard deviation $\sigma$ has a monotonic relationship with entropy, such that maximizing the entropy often drives $\sigma$ to large values, approximating a uniform random policy. Conversely, for a squashed univariate normal distribution, entropy increases with $\sigma$ only up to a certain threshold, beyond which it begins to decrease (see Fig. 2).

Incremental methods can be particularly prone to issues stemming from large and noisy gradients. While off-policy batch methods such as SAC and TD3 benefit from many compute-intensive gradient updates, which effectively smooth out noisy gradients, incremental methods require alternative strategies to manage large gradient updates. Hence, we focus on additional incremental normalization and scaling methods that help stabilize the learning process. These techniques can be seamlessly incorporated into our algorithm with minimal computational overhead. Sec. 5 provides an in-depth discussion that motivates and comprehensively analyzes the impact of the normalization and scaling techniques used in our proposed algorithm.

Stable learning in AVG is achieved by normalizing inputs and hidden unit activations, as well as scaling the temporal difference error. Below, we outline three normalization and scaling techniques used in AVG (more details in Sec. 5).

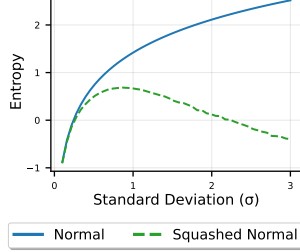

Figure 2: Effect of $\sigma$ on entropy of normal and squashed normal distribution

**Observation normalization** We normalize the observation, which is a commonly used technique in on-policy RL algorithms such as PPO to attain good learning performance. We use an online

algorithm to estimate the sample mean and variance (Welford 1962, See Alg. 2). Sample running mean and variance are effective for stationary and transient distributions, enabling continuous updates that adapt to time-varying characteristics efficiently. In contrast, weighted means emphasize recent observations, making them ideal when recent data points hold greater importance. We use the sample running mean since standard continuous control benchmarks exhibit transient distributions for policies.

**Penultimate Normalization** Bjorck et al. (2022) suggest normalizing features ($\psi_\theta(S)$) of the penultimate layer of a neural network. These features are normalized into a unit vector $\hat{\psi}_\theta(S) = \psi_\theta(S)/\|\psi_\theta(S)\|_2$, with gradients computed through the feature normalization. Unlike layer normalization (Ba et al. 2016), no mean subtraction is performed.

**Scaling Temporal Difference Errors** Schaul et al. (2021) proposed replacing raw temporal difference (TD) errors $\delta_t$ by a scaled version: $\bar{\delta}_t := \delta_t/\sigma_\delta$ where $\sigma_\delta^2 := \mathbb{V}[R] + \mathbb{V}[\gamma]\mathbb{E}[G^2]$. This technique can handle varying episodic return scales across domains, tasks, and stages of learning. It is also algorithm-agnostic and does not require access to the internal states of an agent. In batch RL methods with a replay buffer, $\sigma_\delta$ can be computed offline by aggregating the discounted return from each state across stored episodes. However, in the incremental setting, where past data cannot be reused, this approach is infeasible. Consequently, we only use the cumulative return starting from the episode's initial state (See Alg. 3). We also use sample mean and variance of $R, \gamma$ and $G^2$ to calculate $\sigma_\delta$.

**On the Theory of AVG** In Appendix I, we provide a convergence analysis for the reparameterization gradient estimator, which the AVG estimator (1) builds upon. The analysis fixes errors in the convergence result for deterministic policies from Xiong et al. (2022) and extends it to the general case of reparameterized policies. To the best of our knowledge, this is the first convergence result for model-free methods that use the reparameterization gradient estimator. Furthermore, a detailed discussion of related theoretical results is also included in Appendix A.

---

**Algorithm 2** Normalize (Welford 1962)

**Require:** Input $X, n, \mu, \overline{\mu}$
$\quad n \leftarrow n + 1$
$\quad \delta \leftarrow X - \mu$
$\quad \mu \leftarrow \mu + \delta/n$
$\quad \delta_2 \leftarrow X - \mu$
$\quad \overline{\mu} \leftarrow \overline{\mu} + \delta \cdot \delta_2$
$\quad \sigma \leftarrow \overline{\mu}/n$
$\quad X_{norm} \leftarrow \delta_2/\sigma$
$\quad$ **return** $X_{norm}, n, \mu, \overline{\mu}, \sigma$

---

**Algorithm 3** ScaleTDError

**Require:** Input $R, \gamma, G, \boldsymbol{n}_\delta, \boldsymbol{\mu}_\delta, \overline{\boldsymbol{\mu}}_\delta$
$\quad n_R, n_\gamma, n_G \leftarrow \boldsymbol{n}_\delta; \quad \mu_R, \mu_\gamma, \mu_G \leftarrow \boldsymbol{\mu}_\delta$
$\quad \overline{\mu}_R, \overline{\mu}_\gamma, \overline{\mu}_G \leftarrow \overline{\boldsymbol{\mu}}_\delta \qquad \triangleright \mu_G$: Sample mean of $G^2$
$\quad \_, n_R, \mu_R, \overline{\mu}_R, \sigma_R \leftarrow \texttt{Normalize}(R, n_R, \mu_R, \overline{\mu}_R)$
$\quad \_, n_\gamma, \mu_\gamma, \overline{\mu}_\gamma, \sigma_\gamma \leftarrow \texttt{Normalize}(\gamma, n_\gamma, \mu_\gamma, \overline{\mu}_\gamma)$
$\quad$ **if** G is not $\emptyset$ **then**
$\quad\quad \_, n_G, \mu_G, \overline{\mu}_G, \_ \leftarrow \texttt{Normalize}(G^2, n_G, \mu_G, \overline{\mu}_G)$
$\quad$ **end if**
$\quad$ **if** $n_G > 1$ **then**
$\quad\quad \sigma_\delta \leftarrow \sqrt{\sigma_R^2 + \mu_G \sigma_\gamma^2}$
$\quad$ **else**
$\quad\quad \sigma_\delta \leftarrow 1$
$\quad$ **end if**
$\quad \boldsymbol{n}_\delta \leftarrow [n_R, n_\gamma, n_G]; \quad \boldsymbol{\mu}_\delta \leftarrow [\mu_R, \mu_\gamma, \mu_G]$
$\quad \overline{\boldsymbol{\mu}}_\delta \leftarrow [\overline{\mu}_R, \overline{\mu}_\gamma, \overline{\mu}_G]$
$\quad$ **return** $\sigma_\delta, \boldsymbol{n}_\delta, \boldsymbol{\mu}_\delta, \overline{\boldsymbol{\mu}}_\delta$

---

## 4 AVG on Simulated Benchmark Tasks

In this section, we demonstrate the superior performance of AVG compared to existing incremental learning methods. Specifically, we compare AVG against an existing incremental method — IAC, which has demonstrated strong performance with linear function approximation in real-time learning across both simulated and real-world robot tasks (Degris et al. 2012, Vasan 2017). The implementation details can be found in Appendix E. Additionally, we evaluate AVG against incremental adaptations of SAC and TD3, both of which, like AVG, use RG estimation.

SAC and TD3 rely on large replay buffers to store and replay past experiences, a crucial feature for tackling challenging benchmark tasks. To adapt these batch-based methods to an incremental setting, we set the minibatch and replay buffer size to 1, allowing them to process each experience as it is encountered. We refer to these incremental variants as *SAC-1* and *TD3-1*, respectively. We use off-the-shelf implementations of TD3 and SAC provided by CleanRL (Huang et al. 2022b). The choice of hyper-parameters and full learning curves can be found in the Appendix F.

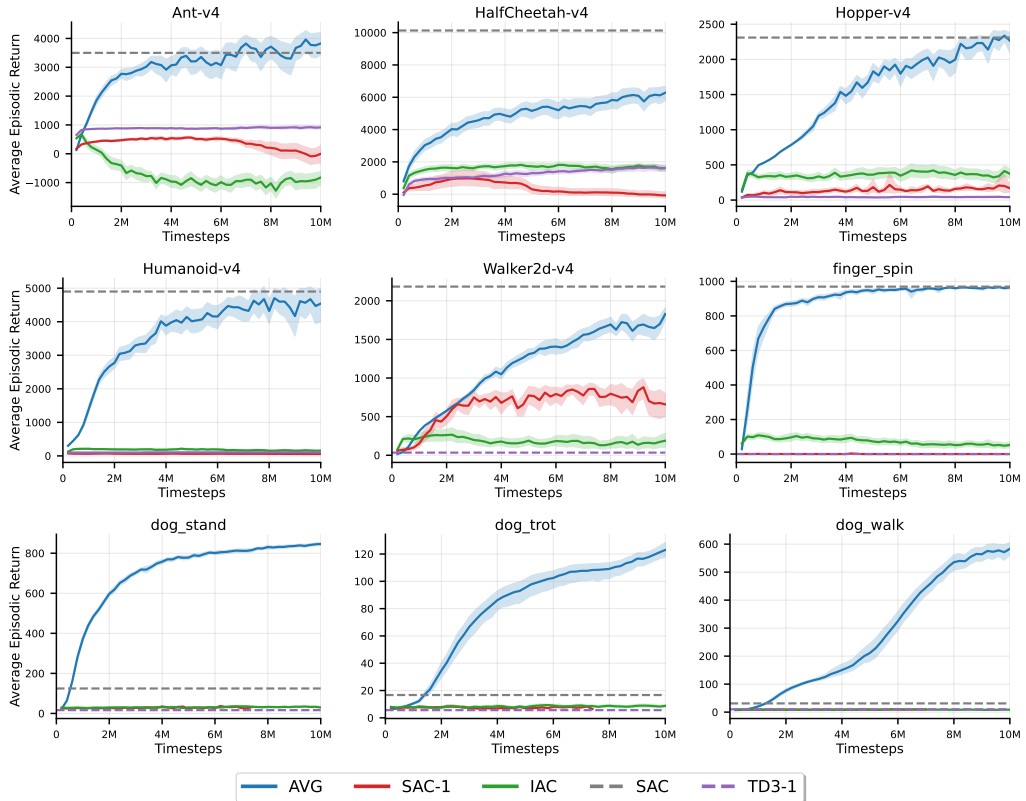

Figure 3: AVG on Gymnasium and DeepMind Control Suite tasks. Each solid learning curve is an average of 30 independent runs. The shaded regions represent a 95% confidence interval of the bootstrap distribution. Note that *SAC* refers to SAC with a replay buffer size of $1M$. The corresponding dashed line represents the mean performance over the final 10K steps of training.

In Figure 3, we present the learning performance of AVG in comparison to IAC, SAC-1, and TD3-1. For reference, we also include the final performance of SAC with large replay buffers and default parameters, trained for $1M$ timesteps, indicated by the gray dashed line (referred to as *SAC*). Notably, AVG is the only incremental algorithm that learns effectively, achieving performance comparable to SAC in Gymnasium (Towers et al. 2023) environments and surpassing it in the Dog benchmarks from DeepMind Control Suite (Tassa et al. 2018). Nauman et al. (2024) suggests that non-default regularization, such as layer normalization is essential for SAC to perform well in the Dog domain.

To optimize the hyperparameters for each method—AVG, IAC, SAC-1, and TD3-1, we conducted a random search, which is more efficient for high-dimensional search spaces than grid search (Bergstra and Bengio 2012). We evaluated 300 different hyperparameter configurations, each trained with 10 random seeds for $2M$ timesteps on five challenging continuous control environments: *Ant-v4, Hopper-v4, HalfCheetah-v4, Humanoid-v4* and *Walker2d-v4*. Each configuration was ranked based on its average undiscounted return per run, with the top-performing configuration selected for each environment. Using the best configuration, we then conducted longer training runs of 10 million timesteps with 30 random seeds.

Sparse reward environments can present additional challenges, often increasing both the difficulty and the time required for learning (Vasan et al. 2024). Hence, we also evaluate our algorithms on sparse reward environments from the DeepMind Control Suite. We use one unique hyper-parameter configuration per algorithm across four environments: *finger_spin, dog_stand, dog_walk, dog_trot* (see Fig. 3). Further details are provided in Appendix F.4.

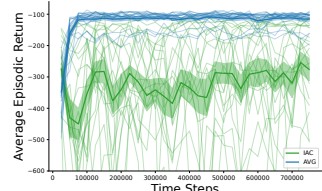

Figure 4: AVG and IAC on the Visual Reacher task

**Learning From Pixels** We use the visual reacher task to ensure that AVG can be used with visual RL. In this task, the agent uses vision and proprioception to reach a goal. As shown in Fig. 4, AVG consistently outperforms IAC, which exhibits high variance and struggles to learn. Task details are provided in Appendix B.3.

# 5 Stabilizing Incremental Policy Gradient Methods

In this section, we first highlight some issues with incremental policy gradient methods, which arise from the large and noisy gradients inherent to the setting. We perform a comprehensive ablation study to assess the effects of observation normalization, penultimate normalization, and TD error scaling—individually and in combination—on the performance of AVG. Additionally, we demonstrate how other incremental methods, such as IAC and SAC-1, may also benefit from normalization and scaling.

## 5.1 Instability Without Normalization

Deep RL can suffer from instability, often manifesting as high variance (Bjorck et al. 2022), reduced expressivity of neural networks over time (Nikishin et al. 2022, Sokar et al. 2023), or even a gradual drop in performance (Dohare et al. 2023, Elsayed and Mahmood 2024, Elsayed et al. 2024a), primarily due to the non-stationarity of data streams. Recently, Lyle et al. (2024) identified another common challenge that may induce difficulty in learning: the large regression target scale. For instance, in the Humanoid-v4 task, bootstrapped targets can range from $-20$ to $8000$ during training. Consequently, the critic faces the difficult task of accurately representing values that fluctuate widely across different stages of training. This can lead to excessively large TD errors, destabilizing the learning process.

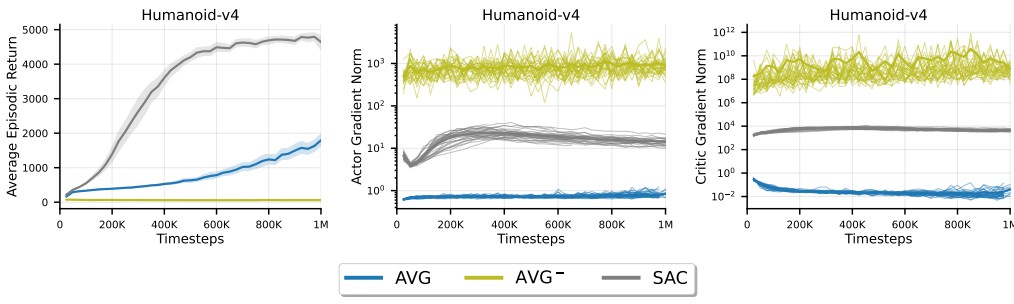

Figure 5: The gradient norm of the critic and actor networks for AVG and SAC, along with their average episodic returns. $AVG^-$ denotes AVG without any normalization or scaling applied. The solid lines represent the average, whereas the light lines represent the values for the individual runs. Note that the y-axis in the plots for actor and critic gradient norms is displayed on a logarithmic scale.

Figure 5 illustrates a failure condition that can arise due to large regression target scale, high variance, and reduced expressivity — challenges that are particularly problematic for incremental methods. Here, we compare a successful SAC training run to a failed AVG run without normalization or scaling techniques (termed $AVG^-$). While batch RL methods like SAC manage large, noisy gradients by smoothing them out through batch updates and improving stability with target Q-networks, incremental methods like AVG are more susceptible to numerically unstable updates, which can lead to failure or divergence in learning. $AVG^-$ exemplifies this issue by demonstrating excessively large gradient norms, particularly in the critic network, resulting in erratic gradients that hinder learning.

Building on these insights, we *hypothesize* that stable learning in AVG can be achieved by balancing update magnitudes across time steps and episodes, reducing the influence of outlier experiences. This can be partly accomplished by centering and scaling the inputs, normalizing the hidden unit activations, and scaling the TD errors. Andrychowicz et al. (2021) show that appropriately scaling the observations can help improve performance, likely since it helps improve learning dynamics (Sutton 1988, Schraudolph 2002, LeCun et al. 2002). Scaling both the targets (e.g., by scaling the rewards, Engstrom et al. 2019) and the observations (e.g., normalization, Andrychowicz et al. 2021) is a well-established strategy that has shown success and is incorporated into widely used algorithms such as PPO (Schulman et al. 2017), helping improve its performance and stability (Rao et al. 2020, Huang et al. 2022a).

## 5.2 Disentangling the Effects of Normalization and Scaling

A combination of three techniques consistently achieves good performance for AVG: 1) TD error scaling (Schaul et al. 2021) to resolve the issue of large bootstrapped target scale (termed *scaled_td*, 2) observation normalization to maintain good learning dynamics (termed *norm_obs*, and 3) penultimate

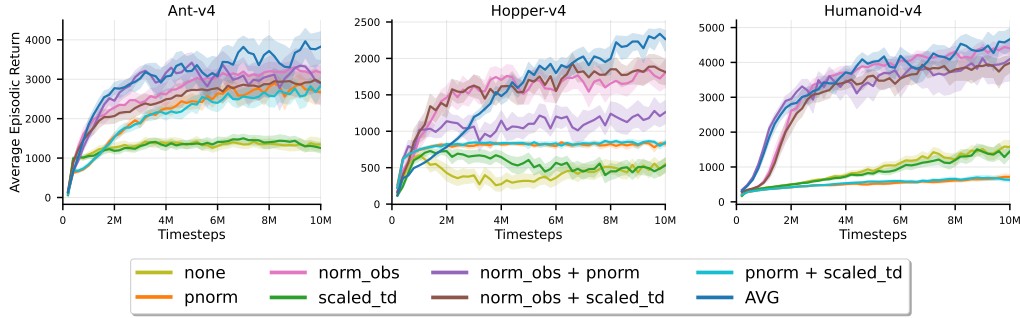

Figure 6: Ablation study of normalization and scaling techniques used with AVG. We plot the learning curves of the best hyperparameter configurations for each task variant. Each solid learning curve is an average of 30 independent runs. The shaded regions represent a 95% confidence interval.

normalization to reduce instability and improve plasticity (termed *pnorm*, Bjorck et al. 2022), similar to layer normalization (Lyle et al. 2023). We selected Welford's online algorithm for normalizing observations due to its unbiased nature and its ability to maintain statistics across the entire data stream. In preliminary experiments, weighted methods that favored more recent samples did not perform well. Schaul et al. (2021) illustrate the risks associated with clipping or normalizing rewards, which led us to adopt their straightforward approach of scaling the temporal difference error with a multiplicative factor. Additionally, we favored pnorm over layer normalization since it performed better empirically in our experiments (see Fig. 13, App. B.2). It is worth noting that alternative normalization techniques could potentially achieve similar, if not superior, outcomes. Our focus here is to emphasize the importance of normalization and scaling issues and propose easy-to-use solutions.

We conduct an ablation study to evaluate the impact of the three techniques—norm_obs, pnorm and scaled_td—on the performance of AVG. We assess these techniques both individually and in combination, resulting in a total of 8 variants. The learning curves for the best seed obtained via our random search procedure (detailed in App. F.4) for each variant are shown in Fig. 6. The combination of all three techniques achieves the best overall performance.

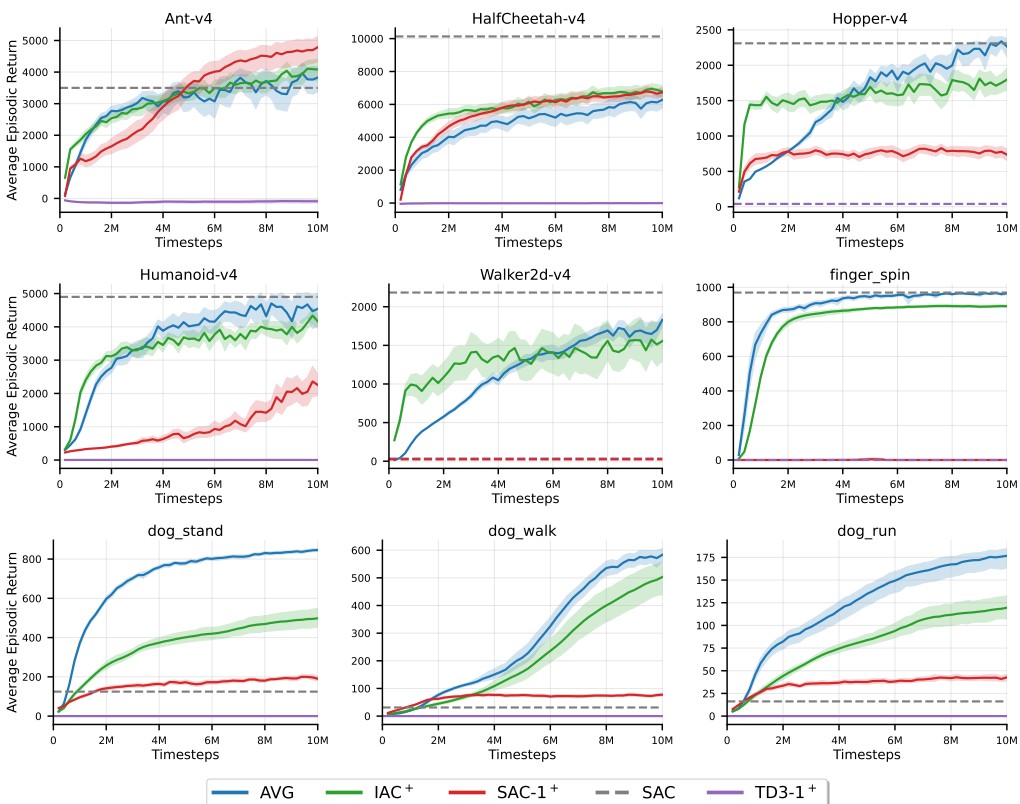

Figure 7: Impact of normalization and scaling on IAC, SAC-1 and TD3-1. Suffix "+" denotes each algorithm plus normalization and scaling.

In Fig. 7, we assess the impact of our proposed normalization and scaling techniques on IAC, SAC-1 and TD3-1. While IAC$^+$ performs in a mostly comparable manner to AVG, SAC-1$^+$ shows inconsistent performance, performing well in only two tasks but failing or even diverging in environments such as Hopper-v4 and Walker2d-v4. TD3-1$^+$ fails to learn in all environments.

## 5.3 AVG with Target Q-Networks

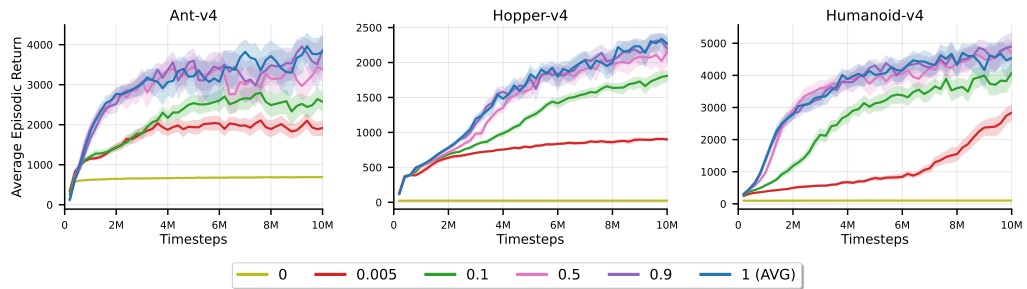

Figure 8: Impact of target Q network on AVG for different values of $\tau$, which represents the Polyak averaging coefficient. Here, $\tau = 0$ corresponds to a fixed target network, and $\tau = 1$ indicates that the current Q-network and the target network are identical, that is, not using a target network.

Target networks are commonly used in off-policy batch methods to stabilize learning (Mnih et al. 2015). By using a separate network that is updated less frequently, target networks introduce a delay in the propagation of value estimates. This delay can be advantageous in batch methods with large replay buffers, as it helps maintain a more stable target (Lillicrap et al. 2015, Fujimoto et al. 2018). However, this delayed update can slow down learning in online RL (Kim et al. 2019).

In Figure 8, we evaluate the impact of using target Q-networks with AVG. Similar to SAC, we use Polyak averaging to update the target Q-network: $\phi_{\text{target}} = (1 - \tau) \cdot \phi_{\text{target}} + \tau \cdot \phi$. We run an experiment varying $\tau$ between $[0, 1]$, where $\tau = 0$ denotes a fixed target network and $\tau = 1$ implies the target network is identical to the current Q-network. We detail the pseudocode in Appendix C (see Alg. 4). The results show no benefit to using target networks, with only large values of $\tau$ performing comparably to AVG. Additionally, removing target networks reduces memory usage and simplifies the implementation of our algorithm.

## 6 AVG with Resource-Constrained Robot Learning

On-device learning enables mobile robots to continuously improve, adapt to new data, and handle unforeseen situations, which is crucial for tasks like autonomous navigation and object recognition. Commercial robots, such as the iRobot Roomba, often use onboard devices with limited memory, ranging from microcontrollers with kilobytes of memory to more powerful edge devices like the Jetson Nano 4GB. Leveraging these onboard edge devices can reduce the need for constant server communication, enhancing reliability in areas with limited

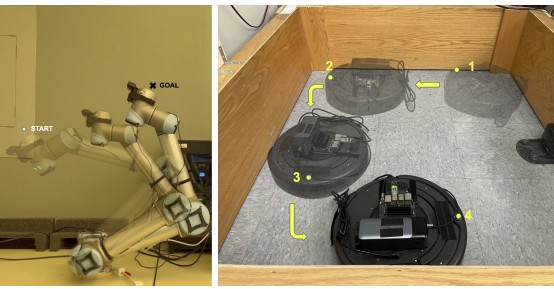

(a) UR-Reacher-2          (b) Create-Mover

Figure 9: Robot Tasks

connectivity. Storing large replay buffers on these devices is infeasible, necessitating computationally efficient, incremental algorithms.

To demonstrate the effectiveness of our proposed AVG algorithm for on-device incremental deep RL, we utilize the *UR-Reacher-2* and *Create-Mover* tasks, as developed by Mahmood et al. (2018). We use two robots: UR5 robotic arm and iRobot Create 2, a hobbyist version of Roomba. In the UR-Reacher-2 task, the agent aims to reach arbitrary target positions on a 2D plane (Fig. 9a). This

task is a real-world adaptation of the Mujoco Reacher task. In the Create-Mover task, the agent's goal is to move the robot forward as fast as possible within an enclosed arena. A representative image of the desired behavior is shown in Fig. 9b. Each run requires slightly over two hours of robot experience time on both robots. In our learning curves (see Fig. 10), the dark lines represent the average over five runs for AVG, whereas the light lines represent the values for the individual runs. Details of the setup can be found in Appendix H.

The performance of AVG and resource-constrained SAC on UR-Reacher-2 is shown in Fig. 10 (top). We term the resource-constrained variants of SAC as *SAC-1* and *SAC-100*, where the suffix indicates both the replay buffer capacity and mini-batch size used during training. Note that SAC-1 is incremental, but SAC-100 is still a batch method with limited memory resources. In these experiments, both SAC-100 and SAC-1 struggle significantly, failing to learn under the imposed memory limitations. In contrast, AVG demonstrates robust performance, efficiently utilizing limited memory to achieve fast and superior learning.

On the mobile robot task Create-Mover, the learning system is limited to onboard computation using a Jetson Nano 4GB. This introduces additional compute constraints in terms of action sampling time and learning update time. Our implementation requires $5ms$ to sample an action for both AVG and SAC-1. On the other hand, for learning updates, AVG requires only about $37ms$ per update, compared to SAC-1's $67ms$. A batch update for SAC would exceed the action cycle time ($150ms$) for Create-Mover. Hence, we compare AVG only against SAC-1. The learning curves on the Create-Mover task in Fig. 10 (bottom) clearly show AVG's superior performance, while SAC-1 fails to learn any meaningful policy. This highlights AVG's efficiency

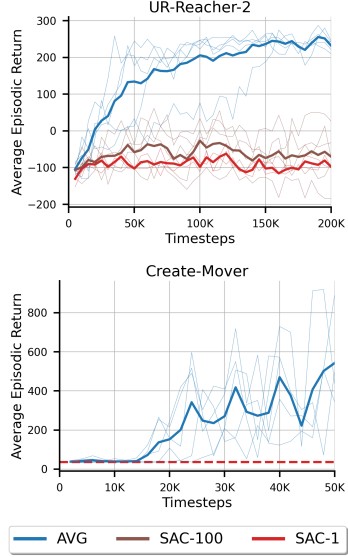

Figure 10: Learning curves on Real Robot Tasks

and suitability for real-time learning in resource-constrained environments. Our work demonstrates for the first time effective real-robot learning with incremental deep reinforcement learning methods[3].

# 7 Conclusion

This work revives incremental policy gradient methods for deep RL and offers significant computational advantages over standard batch methods for onboard robotic applications. We introduced a novel incremental algorithm called Action Value Gradient (AVG) and demonstrated its ability to consistently outperform other incremental and resource-constrained batch methods across a range of benchmark tasks. Crucially, we showed how normalization and scaling techniques enable AVG to achieve robust learning performance even on challenging high-dimensional control problems. Finally, we presented the first successful application of an incremental deep RL method learning control policies from scratch directly on physical robots—a robotic manipulator and a mobile robot. Overall, our proposed AVG algorithm opens up new possibilities for deploying deep RL with limited onboard computational resources of robots, enabling lifelong learning and adaptation in the real world.

**Limitations and Future Work** The main limitation of our approach is low sample efficiency compared to batch methods. Developing AVG with eligibility traces (Singh and Sutton 1996, van Hasselt et al. 2021) is a natural future direction to generalize our one-step AVG and possibly improve its sample efficiency. We also find that AVG can be sensitive to the choice of hyper-parameters. A valuable extension would be stabilizing the algorithm to perform well across environments using the same hyper-parameters. Our work is limited to continuous action space, but it can also be extended to discrete action spaces following Jang et al. (2017), which we leave to future work. Additionally, AVG omits discounting in the state distribution, which is common and further biases the update but can be addressed with the correction proposed by Che et al. (2023). Finally, we acknowledge a concurrent work by Elsayed et al. (2024b), which stabilizes existing incremental methods like AC($\lambda$) and Q($\lambda$), except for reparameterization policy gradient methods. The robustness of AVG may potentially improve by replacing Adam with an optimizer for adaptive step sizes proposed in that work.

---

[3]Video Demo: `https://drive.google.com/drive/folders/183avabM6WsXlakMQn7qYd_F--HJkxBd0`

**Societal Impact** Our paper presents academic findings, but the proposed algorithm offers new opportunities for deploying deep reinforcement learning on robots with limited computational resources. This enables lifelong learning and real-world adaptation, advancing the development of more capable autonomous agents. While our contributions themselves do not cause negative societal effects, we advise the community to reflect on possible consequences as they expand upon our research.

**Acknowledgements** We thank all reviewers for their insightful comments and suggested experiments, which strengthened both the content and presentation of our paper. We would also like to thank Shibhansh Dohare, Kris De Asis, Homayoon Farrahi, Varshini Prakash, and Shivam Garg for their helpful discussions. We are also appreciative of the computing resources provided by the Digital Research Alliance of Canada and the financial support from the CCAI Chairs program, the RLAI laboratory, Amii, and NSERC of Canada.

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

# A Theoretical Foundations

## A.1 Reparameterization Policy Gradient Theorem

Please refer to the Theorems and Proofs section of Lan et al. (2022) for detailed proofs. We only provide a short proof sketch for reference.

**Theorem 1** (Reparameterization Policy Gradient Theorem). *Given an MDP and a policy objective $J(\theta) \doteq \int d_0(s)v_{\pi_\theta}(s)ds$. The reparameterization policy gradient is given as*

$$\nabla_\theta J(\theta) = \mathbb{E}_{S \sim d_{\pi,\gamma}, A \sim \pi_\theta}\left[\nabla_\theta f_\theta(\xi; S)|_{\xi = h_\theta(A;S)}\nabla_A q_{\pi_\theta}(S, A)\right].$$

*Proof.*

$$\nabla_\theta J(\theta) = \nabla_\theta \int d_0(s)v_{\pi_\theta}(s)ds$$

$$= \nabla_\theta \int d_0(s)\left(\int d_{\pi,\gamma}(s)\pi_\theta(a|s)q_{\pi_\theta}(s, a)da\ ds\right)ds$$

$$= \nabla_\theta \int d_0(s)\left(\int d_{\pi,\gamma}(s)p(\xi)q_{\pi_\theta}(s, f_\theta(\xi; s))d\xi\ ds\right)ds \qquad \text{(by reparameterization)}$$

$$= \int d_0(s)\left(\int d_{\pi,\gamma}(s)p(\xi)\nabla_\theta q_{\pi_\theta}(s, f_\theta(\xi; s))d\xi\ ds\right)ds$$

$$= \int d_0(s)\left(\int d_{\pi,\gamma}(s)p(\xi)\nabla_\theta f_\theta(\xi; s)\nabla_a q_{\pi_\theta}(s, a)|_{a = f_\theta(\xi; s)}d\xi\ ds\right)ds \quad \text{(using chain rule)}$$

$$\propto \int d_{\pi,\gamma}(s)p(\xi)\nabla_\theta f_\theta(\xi; s)\nabla_a q_{\pi_\theta}(s, a)|_{a = f_\theta(\xi; s)}d\xi\ ds$$

$$= \int d_{\pi,\gamma}(s)\pi_\theta(a|s)\nabla_\theta f_\theta(\xi; s)|_{\xi = h_\theta(a; s)}\nabla_a q_{\pi_\theta}(s, a)da\ ds \qquad \text{(by back substitution)}$$

$$= \mathbb{E}_{S \sim d_{\pi,\gamma}, A \sim \pi_\theta}\left[\nabla_\theta f_\theta(\xi; S)|_{\xi = h_\theta(A; S)}\nabla_A q_{\pi_\theta}(S, A)\right].$$

$\square$

## A.2 Action Value Gradient Theorem

**Theorem 2** (Action Value Gradient Theorem). *Given an MDP and a policy objective $J(\theta) \doteq \int d_0(s)v_{\pi_\theta}^{Ent}(s)ds$. The action value gradient is given as*

$$\nabla_\theta J(\theta) = \mathbb{E}_{S \sim d_{\pi,\gamma}, A \sim \pi_\theta}\left[\nabla_\theta f_\theta(\xi; S)|_{\xi = h_\theta(A; S)}\nabla_A(q_{\pi_\theta}(S, A) - \eta \log \pi_\theta(A|S))\right].$$

*Proof.*

$$\nabla_\theta J(\theta) = \nabla_\theta \int d_0(s)v_{\pi_\theta}^{\text{Ent}}(s)ds$$

$$= \nabla_\theta \int d_0(s)\left(\int d_{\pi,\gamma}(s)\pi_\theta(a|s)(q_{\pi_\theta}(s, a) - \eta\mathcal{H}(\cdot|s))da\ ds\right)ds$$

$$= \nabla_\theta \int d_0(s)\left(\int d_{\pi,\gamma}(s)\pi_\theta(a|s)(q_{\pi_\theta}(s, a) - \eta\log\pi(a|s))da\ ds\right)ds$$

$$= \nabla_\theta \int d_0(s)\left(\int d_{\pi,\gamma}(s)p(\xi)(q_{\pi_\theta}(s, f_\theta(\xi; s)) - \eta\log\pi(f_\theta(\xi; s)|s))d\xi\ ds\right)ds$$

$$= \int d_0(s)\left(\int d_{\pi,\gamma}(s)p(\xi)\nabla_\theta\left(q_{\pi_\theta}(s, f_\theta(\xi; s)) - \eta\log\pi(f_\theta(\xi; s)|s)\right)d\xi\ ds\right)ds$$

$$= \int d_0(s)\left(\int d_{\pi,\gamma}(s)p(\xi)\nabla_\theta f_\theta(\xi; s)\nabla_a(q_{\pi_\theta}(s, a) - \eta\log\pi(f_\theta(\xi; s)|s))|_{a = f_\theta(\xi; s)}d\xi\ ds\right)ds$$

$$\propto \int d_{\pi,\gamma}(s)p(\xi)\nabla_\theta f_\theta(\xi; s)\nabla_a(q_{\pi_\theta}(s, a) - \eta\log\pi(f_\theta(\xi; s)|s))|_{a = f_\theta(\xi; s)}d\xi\ ds$$

$$= \int d_{\pi,\gamma}(s)\pi_\theta(a|s)(\nabla_\theta f_\theta(\xi;s)|_{\xi=h_\theta(a;s)}\nabla_a(q_{\pi_\theta}(s,a) - \eta\log\pi(a|s))da\ ds$$

$$= \mathbb{E}_{S\sim d_{\pi,\gamma}, A\sim\pi_\theta}\left[\nabla_\theta f_\theta(\xi;S)|_{\xi=h_\theta(A;S)}\nabla_A(q_{\pi_\theta}(S,A) - \eta\log\pi(A|S))\right].$$

We get the third line since $\forall s$, the term $\log\pi(A|s)$ is unbiased estimate of $\mathcal{H}(\cdot|s)$ so we can write $\mathcal{H}(\cdot|s) = \mathbb{E}[\log\pi(A|s)], \forall s$.

$\square$

## A.3   Related Theoretical Works

We review relevant theoretical works on the convergence of actor-critic algorithms. To the best of our knowledge, there is no existing proof for the exact action value gradient (AVG) algorithm used in our paper. However, there are studies of algorithms similar to ours that provide some theoretical justification.

We begin by considering the case without entropy regularization. Xiong et al. (2022) examine the convergence of deterministic policy gradient (DPG; Silver et al. 2014) algorithms. Their online version of DPG employs i.i.d. samples of states from the stationary distribution, which differs from the single stream of experience examined in our study. In addition, DPG uses a deterministic policy with a fixed exploration noise distribution, whereas AVG also learns the exploration parameter. Nevertheless, this work is one of the closest to ours, as it uses the reparameterized gradient estimator in the update. Besides, Bhatnagar et al. (2007) provide convergence guarantees for incremental actor-critic algorithms. However, their results are based on the likelihood-ratio estimator, and thus applicable to incremental actor critic but not AVG.

When considering the entropy-regularized objective, most studies assume the presence of the true gradient (Mei et al. 2020, Cen et al. 2022), with the exception of Ding et al. (2021), which provides an asymptotic convergence guarantee to stationary points for entropy-regularized actor-critic algorithms. However, their algorithm differs from AVG in two aspects: First, the samples used in their update are from the discounted stationary distribution; second, they also use the likelihood-ratio estimator. Nonetheless, their work offers valuable insights into the theoretical underpinnings of the entropy-regularized objective.

Despite these differences, it is reasonable to hypothesize that our algorithm converges, given the convergence guarantees for algorithms closely related to ours. The techniques from these works may be useful for demonstrating the convergence of our algorithm. For example, we can extend the convergence analysis for deterministic policies in Xiong et al. (2022) to the general case of reparameterized policies, as shown in Appendix I.

# B    AVG Design Choices

**Orthogonal Initialization** helps improve the training stability and convergence speed of neural networks by ensuring that the weight matrix has orthogonal properties, thereby preserving the variance of the input through the layers (Saxe et al. 2013).

**Squashed Normal Policy** SAC utilizes a *squashed Normal*, where the unbounded samples from a Normal distribution are passed through the `tanh` function to obtain bounded actions in the range $[-1, 1] : A_\theta = f_\theta(\epsilon; S) = \mathtt{tanh}(\mu_\theta(S) + \sigma_\theta(S)\epsilon)$ where $\epsilon \sim \mathcal{N}(0, 1)$. This parameterization is useful for entropy-regularized RL objectives, which maximizes the return based on the maximum-entropy formulation. With an unbounded Normal policy, the standard deviation $\sigma$ has a linear relationship with entropy. Hence, learning to maximize entropy can often result in very large values of $\sigma$, potentially leading to behavior resembling a uniform random policy. In contrast, for a univariate squashed Normal with zero mean, increasing $\sigma$ does not continuously maximize the entropy; it decreases after a certain threshold.

**Entropy Regularization** Given that batch methods such as SAC benefit from entropy regularization, we consider variants of AVG with and without entropy regularization. There are two types of entropy terms that can be added to the actor, and critic updates: 1) *distribution entropy*: $\mathcal{H}(\pi(\cdot|S))$, and 2) *sample entropy*: $-\log(\pi(A|S))$. We use sample entropy as our final choice in Algorithm 1, which utilizes sample entropy for the regularization of both the actor and Q-network.

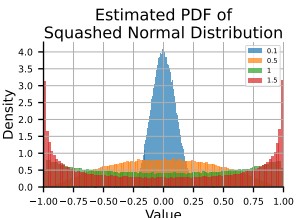

Figure 11: Squashed Normal Distribution PDF

Simply increasing $\sigma$ does not maximize the entropy of a univariate squashed Normal with zero mean (see Fig. 2). Increasing $\sigma$ results in the probability density function (PDF) of a squashed Normal concentrating at the edges (Fig. 11), resembling bang-bang control (Seyde et al. 2021).

## B.1    Relative Performance of Different Hyperparameter Configurations in Random Search

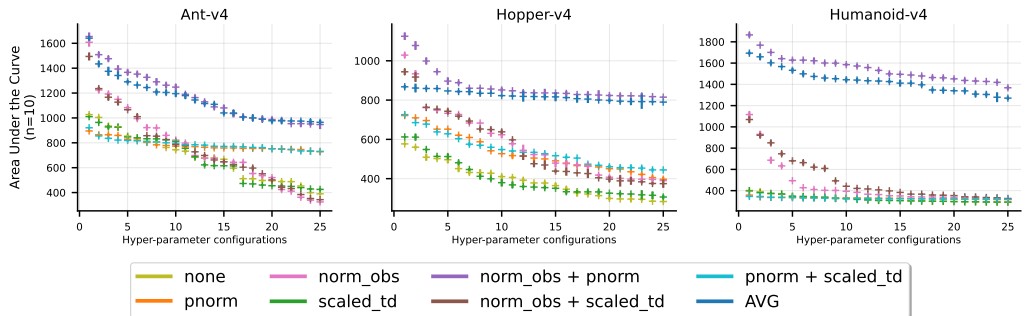

Figure 12: Hyperparameter Evaluation via Random Search. Scatter plot of the performance of the best 25 out of 300 unique hyper-parameter configurations. Note that the y-axis represents the area under the curve for 2M timesteps, not an evaluation of the final policy for 10M timesteps.

We conduct an ablation study to evaluate the impact of these techniques on the performance of AVG. We assess these techniques both individually and in combination, resulting in a total of 8 variants. AVG, without any normalization or scaling techniques, serves as the *baseline*. We test 300 unique hyper-parameter configurations for each variant, trained for $2M$ timesteps with 10 random seeds on the Ant-v4, Hopper-v4 and Humanoid-v4 environments. We then calculate the average undiscounted return for each run (i.e., the area under the curve [AUC]) and average the AUC across all 10 seeds. We plot the top 30 hyper-parameter configurations in Fig. 12 as a scatter plot, ranked in descending order from highest to lowest mean AUC. Each plot point represents the mean AUC, and the thin lines denote the standard error. Note that a point for a hyper-parameter configuration is plotted only if the configuration runs without diverging on all 10 random seeds. These plots can be indicative of which variant would obtain the highest average episodic return when trained for longer and the robustness of the variants to the choice of hyper-parameters.

## B.2  The Effect of Other Network Normalization Techniques

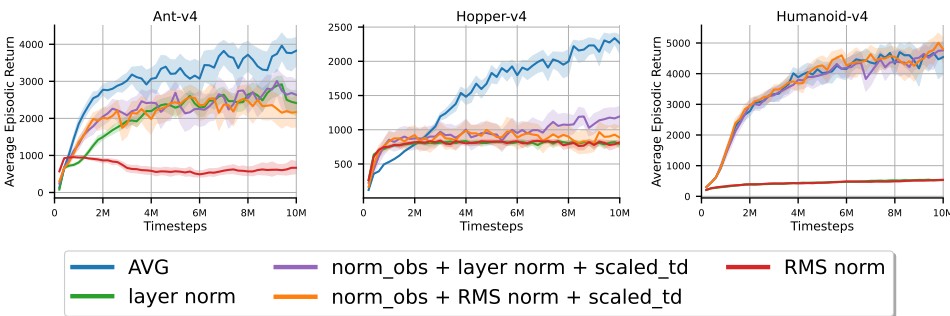

Figure 13: Comparing different neural network feature normalizations — penultimate normalization (pnorm) against layer normalization (layer norm) and RMS normalization (RMS norm)

When comparing different neural network feature normalizations, penultimate normalization (pnorm), layer normalization (layer norm), and RMS normalization (RMS norm) each offer unique advantages depending on the architecture and task. Layer normalization (Ba et al. 2016) is widely used for stabilizing hidden layer pre-activations by normalizing across features within each layer, which helps models converge consistently across a variety of tasks. Root mean square layer normalization, on the other hand, normalizes based on the root mean square of pre-activations, providing stability without fully normalizing the mean, which can be beneficial in reducing variance across diverse input patterns (Zhang and Sennrich 2019). Penultimate normlazation focuses on the penultimate layer activations, normalizing just before the final layer, which allows it to maintain high-quality feature representations critical for downstream performance (Bjorck et al. 2022).

Figure 13 indicates that AVG, which uses penultimate normalization, consistently outperform those with other normalizations. We use the random search procedure described earlier to identify the hyperparameter configuration for each variant.

## B.3  Vision-Based Learning using AVG

This task involves moving a two-degree-of-freedom (DoF) planar arm's fingertip to a random spherical target on a 2D plane. It includes two sub-tasks (easy, hard), that vary in target and fingertip sizes. It is an adaptation of the dm_control reacher Tassa et al. (2018).

For the non-visual task, observations include the fingertip's position, speed, and the fingertip-to-target vector. For the visual task, the fingertip-to-target vector is removed, and the agent receives three consecutive stacked images of size $84 \times 84 \times 3$.

The action space consists of torques applied to the two joints, scaled from $[-1, -1]$ to $[1, 1]$. The reward function is modified to give $-1$ per step, encouraging shorter episodes. After each timeout, the fingertip is reset to a random location while the target remains unchanged. Episodes terminate when the fingertip reaches the target within its size, and upon termination, the agent is reset and a new target is randomly generated for the next episode.

**Convolutional Neural Network Architecture** Our convolutional neural network (CNN) architecture comprises four convolutional layers, followed by a combination of a Spatial Softmax layer and proprioception information. The convolutional layers have 32 output channels and 3x3 kernels, with stride of two for the first three layers and one for the last layer. After these convolutional layers, we use spatial-softmax (Levine et al. 2016) to convert the encoding vector into soft coordinates to track the target more precisely. Additionally, proprioception information is concatenated with the spatial softmax features. The exact number of parameters depends on the input data size and task-specific requirements. The two MLP layers have 512 hidden units each. All the layers except the final output layer use ReLU activation.

# C   AVG with Target Q Networks

---

**Algorithm 4** Action Value Gradient With Target Q-Networks

---

**Initialize** $\gamma, \eta, \alpha_\pi, \alpha_Q, \tau$
$\theta, \phi$ with penultimate normalization
$n \leftarrow 0, \mu \leftarrow 0, \overline{\mu} \leftarrow 0$
$\boldsymbol{n}_\delta \leftarrow [0,0,0], \boldsymbol{\mu}_\delta \leftarrow [0,0,0], \overline{\boldsymbol{\mu}}_\delta \leftarrow [0,0,0]$
$\bar{\phi} \leftarrow \phi$ (target Q-network)
**for** however many episodes **do**
    Initialize S (first state of the episode)
    $S, n, \mu, \overline{\mu}, \_ \leftarrow \texttt{Normalize}(S, n, \mu, \overline{\mu})$
    $G \leftarrow 0$
    **while** S is not terminal **do**
        $A_\theta = f_\theta(\epsilon; S)$ where $\epsilon \sim \mathcal{N}(0,1)$
        Take action $A_\theta$, observe $S', R$
        $S', n, \mu, \overline{\mu}, \_ \leftarrow \texttt{Normalize}(S', n, \mu, \overline{\mu})$
        $\sigma_\delta, \boldsymbol{n}_\delta, \boldsymbol{\mu}_\delta, \overline{\boldsymbol{\mu}}_\delta \leftarrow \texttt{ScaleTDError}(R, \gamma, \emptyset, \boldsymbol{n}_\delta, \boldsymbol{\mu}_\delta, \overline{\boldsymbol{\mu}}_\delta)$
        $G \leftarrow G + R$
        $A' \sim \pi_\theta(\cdot|S')$
        $\delta \leftarrow R + \gamma(Q_{\bar{\phi}}(S', A') - \eta \log \pi_\theta(A'|S')) - Q_\phi(S, A_\theta)$
        $\delta \leftarrow \delta/\sigma_\delta$
        $\phi \leftarrow \phi - \alpha_Q \delta \, \nabla_\phi \, Q_\phi(S, a)|_{a=A_\theta}$
        $\theta \leftarrow \theta + \alpha_\pi \nabla_\theta(Q_\phi(S, A_\theta) - \eta \log \pi_\theta(A_\theta|S))$
        $\bar{\phi} \leftarrow (1-\tau)\bar{\phi} + \tau\phi$
        $S \leftarrow S'$
    **end while**
    $\sigma_\delta, \boldsymbol{n}_\delta, \boldsymbol{\mu}_\delta, \overline{\boldsymbol{\mu}}_\delta \leftarrow \texttt{ScaleTDError}(R, 0, G, \boldsymbol{n}_\delta, \boldsymbol{\mu}_\delta, \overline{\boldsymbol{\mu}}_\delta)$
**end for**

---

# D   Incremental Soft Actor Critic (SAC-1)

---

**Algorithm 5** Incremental SAC (SAC-1)

---

1: **Initialize** policy parameters $\theta$, Q-function parameters $\phi_1$, $\phi_2$, discount factor $\gamma$, polyak averaging coefficient $\rho$ and learnable entropy coefficient $\alpha_\eta$

2: **Initialize** target Q-network parameters $\bar{\phi}_1 \leftarrow \phi_1, \bar{\phi}_2 \leftarrow \phi_2$

3: **for** however many episodes **do**

4:     Initialize S (first state of the episode)

5:     **while** S is not terminal **do**

6:         Sample action $A \sim \pi_\theta(\cdot|S)$

7:         Execute action $A$ in the environment

8:         Observe next state $S'$, reward $R$

9:         $A' \sim \pi_\theta(\cdot|S')$

10:        **for** $i \in \{1, 2\}$ **do**

11:           $\delta \leftarrow R + \gamma(Q_{\bar{\phi}_i}(S', A') - \eta \log \pi_\theta(A'|S')) - Q_{\phi_i}(S, A)$

12:           $\phi \leftarrow \phi - \alpha_Q \delta \nabla_\phi Q_\phi(S, A)$                  ▷ Critic update

13:        **end for**

14:        $X_\theta \sim f_\theta(\xi; S)$ where $\xi \sim \mathcal{N}(0, 1)$

15:        $\theta \leftarrow \theta + \nabla_\theta \left(\min_{i=1,2} Q_{\phi_i}(S, X_\theta) - \alpha \log \pi_\theta(X_\theta|S)\right)|_{\alpha=\alpha_\eta}$    ▷ Actor update

16:        $\eta \leftarrow \eta - \nabla_\eta \alpha_\eta (- \log \pi_\theta(X|S) - target\_entropy)|_{X=X_\theta}$

17:        **for** $i \in \{1, 2\}$ **do**

18:           $\bar{\phi}_i \leftarrow \rho\phi_i + (1 - \rho)\bar{\phi}_i$                 ▷ Update target networks

19:        **end for**

20:        $S \leftarrow S'$

21:     **end while**

22: **end for**

---

We outline the components of AVG and SAC for clarity:

| SAC | AVG |
|:---:|:---:|
| 1 actor | 1 actor |
| 2 Q networks (i.e., double Q-learning) | 1 Q network |
| 2 target Q networks | 0 target networks |
| Learned entropy coefficient $\eta$ | Fixed entropy coefficient $\eta$ |
| Replay buffer $\mathcal{B}$ | No buffers |

Table 1: Comparison of SAC and AVG algorithms

In addition, SAC is off-policy, whereas AVG is on-policy. SAC samples an action and stores it in the buffer. Unlike AVG, SAC's action is not reused to update the actor.

# E   Incremental Actor Critic

We consider the one-step actor-critic by Sutton and Barto (2018), where the *actor* (i.e., policy) and *critic* (i.e., value function) are updated incrementally, as new transitions are observed, rather than waiting for complete episodes or batches of data. We also drop the discount correction term in actor updates since it often leads to poor performance empirically (Nota and Thomas 2019).

We also consider an entropy regularization term in the actor and critic objectives to encourage exploration and discourage premature convergence to a deterministic policy (Williams and Peng 1991, Mnih et al. 2016). In the following subsection, we examine both distribution entropy and sample entropy, finding that distribution entropy performs better empirically. The pseudocode for our implementation of IAC is detailed in Alg. 6.

---

**Algorithm 6** Incremental Actor Critic (IAC)

---

**Initialize** $\theta, \phi, \gamma, \eta, \alpha_\pi, \alpha_V$
**for** however many episodes **do**
    Initialize S (first state of the episode)
    **while** S is not terminal **do**
        $A \sim \pi_\theta(\cdot|S)$
        Take action $A$, observe $S', R$
        $\delta \leftarrow R + \gamma V_\phi(S') - V_\phi(S)$
        $\phi \leftarrow \phi + \alpha_V \delta \nabla_\phi V_\phi(S)$
        $\theta \leftarrow \theta + \alpha_\pi \nabla_\theta(\log(\pi_\theta(A|S)) \delta + \eta \mathcal{H}(\pi_\theta(\cdot|S))$
        $S \leftarrow S'$
    **end while**
**end for**

---

## E.1   Ablation Study of IAC: Distribution against Sample Entropy

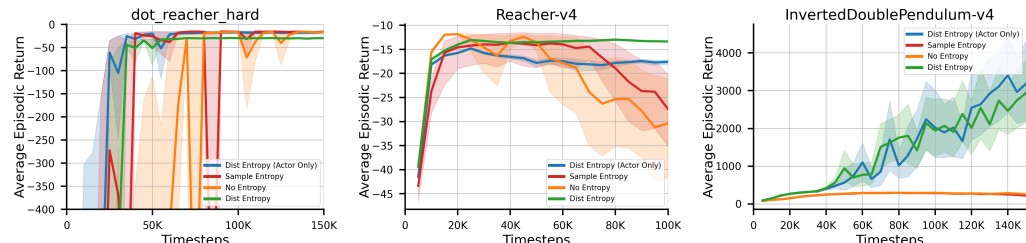

Figure 14: Performance of IAC Variants. Learning curves of the best hyper-parameter configurations found via random search for each task variant. Each solid curve is averaged over 30 independent runs. The shaded regions represent a 95% confidence interval.

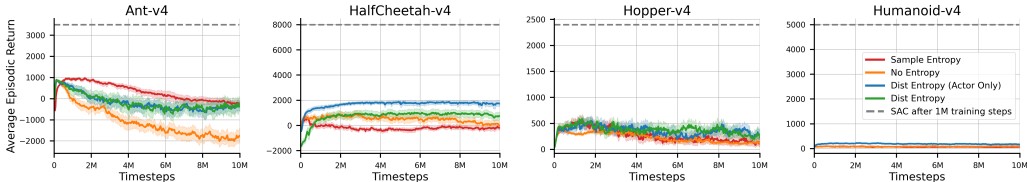

Figure 15: Performance of Incremental Actor Critic (IAC) Variants. Each solid learning curve is an average of 30 independent runs. The shaded regions represent a 95% confidence interval.

## E.2 Impact of Normalization & Scaling on IAC

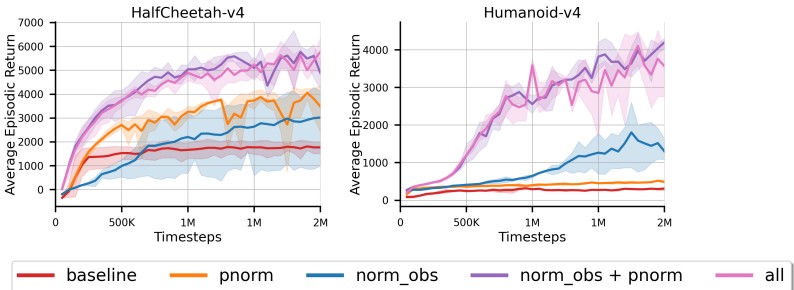

Figure 16: Ablation study of normalization and scaling techniques used with IAC (Algo. 6). We plot the learning curves of the best hyper-parameter configurations for each task variant. Each solid learning curve is an average of 3 independent runs. The shaded regions represent a 95% confidence interval.

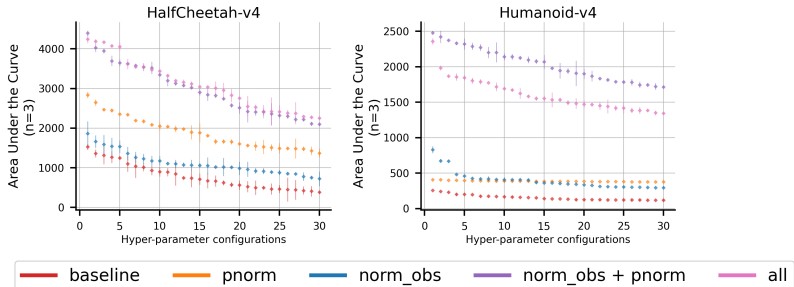

Figure 17: Hyperparameter Evaluation via Random Search. Scatter plot of the performance of the best 30 out of 300 unique hyper-parameter configurations. Note that the y-axis represents the area under the curve, not an evaluation of the final policy.

Our results show that IAC also benefits from normalization and scaling techniques used in AVG (see Fig. 16). The hyperparameter evaluation via random search (Fig. 17) highlights the top 30 configurations out of 300, with the y-axis representing area under the curve rather than final policy performance.

# F Hyper-parameter Settings in Simulation

## F.1 Choice of Hyper-parameters for PPO

Proximal Policy Optimization (PPO) Schulman et al. (2017) introduced PPO, an on-policy policy gradient method. It incorporates proximal optimization ideas to prevent large policy updates, improving stability through its carefully designed surrogate objective.

We use an off-the-shelf implementation of PPO from CleanRL that can be found here: `https://github.com/vwxyzjn/cleanrl/blob/8cbca61360ef98660f149e3d76762350ce613323/cleanrl/ppo_continuous_action.py`

| Parameter | Default Value |
|---|---|
| Update Every | 2048 |
| Minibatch Size | 32 |
| GAE Lambda ($\lambda$) | 0.95 |
| Discount factor ($\gamma$) | 0.99 |
| Num. Optimizer Epochs | 10 |
| Entropy Coefficient | 0 |
| Learning Rate | $3 \times 10^{-4}$ |
| Clip Coefficient ($\epsilon$) | 0.1 |
| Value Loss Coefficient | 0.5 |
| Max Grad Norm | 0.5 |

Table 2: Default parameters for CleanRL PPO implementation.

## F.2 Choice of Hyper-parameters for TD3

Twin Delayed Deep Deterministic Policy Gradient (TD3) Fujimoto et al. (2018) introduced an off-policy algorithm that builds upon DDPG (Lillicrap et al. 2015) known as TD3. Both DDPG and TD3 utilize the reparameterization gradient, albeit for deterministic policies. They made three key modifications that resulted in better performance: (1) using two deep Q-networks to address overestimation bias, (2) delaying updates of the actor-network to reduce per-update error accumulation, and (3) adding noise to the target action used for computing the critic target values.

We use an off-the-shelf implementation of TD3 from CleanRL that can be found here: `https://github.com/vwxyzjn/cleanrl/blob/8cbca61360ef98660f149e3d76762350ce613323/cleanrl/td3_continuous_action.py`

| Parameter | Default Value |
|---|---|
| Replay Buffer Size | 1000000 |
| Minibatch Size | 256 |
| Discount factor ($\gamma$) | 0.99 |
| Policy Noise | 0.2 |
| Exploration Noise | 0.1 |
| Learning Rate | $3 \times 10^{-4}$ |
| Update Every | 2 |
| Noise Clip | 0.5 |
| Learning Starts | 25000 |
| Target Smoothing Coefficient ($\tau$) | 0.005 |

Table 3: Default parameters for CleanRL TD3 implementation.

### F.3 Choice of Hyper-parameters for SAC

**Soft Actor-Critic (SAC)** SAC is an off-policy algorithm which uses the reparametrization gradient along with entropy-augmented rewards (Haarnoja et al. 2018). While TD3 learns a deterministic policy, SAC learns a stochastic policy. TD3 adds noise to the target policy for exploration, whereas SAC's stochastic policy inherently explores by sampling actions from a distribution. We use an adaptation of Vasan et al. (2024) as our baseline SAC implementation.

| Parameter | Default Value |
|---|---|
| Replay Buffer Size | 1000000 |
| Minibatch Size | 256 |
| Discount factor ($\gamma$) | 0.99 |
| Learning Rate | $3 \times 10^{-4}$ |
| Update Actor Every | 1 |
| Update Critic Every | 1 |
| Update Critic Target Every | 1 |
| Learning Starts | 100 |
| Target Smoothing Coefficient ($\tau$) | 0.005 |
| Target Entropy | $|\mathcal{A}|$ |

Table 4: Default parameters for SAC implementation.

### F.4 Hyper-Parameter Optimization Using Random Search

Our random search procedure for hyper-parameter optimization first involves initializing a random number generator (RNG) using unique seed values to ensure reproducibility. Then we use the RNG to sample learning rates for the actor and critic networks, parameters for the Adam optimizer, entropy coefficient, discount factor ($\gamma$) and polyak averaging constant (if applicable). The ranges of hyper-parameter values we use in this experiment are listed in Table 5.

| Hyperparameter | Range |
|---|---|
| `actor_lr` | $10^{[-2,-6]}$ |
| `critic_lr` | $10^{[-2,-6]}$ |
| `Optimizer` | Adam |
| `beta1` | $\{0, 0.9\}$ |
| `beta2` | 0.999 |
| `alpha_lr` | $10^{[-5,0]}$ |
| `gamma` | $\{0.95, 0.97, 0.99, 0.995, 1.0\}$ |
| `critic_tau` | 0.005 if `algo` $\in \{$SAC, TD3$\}$ |
| `NN Activation` | Leaky ReLU |
| `Num. hidden layers` | 2 |
| `Num hidden units` | 256 |
| `Weight initialization` | Orthogonal |

Table 5: Random Search Procedure for Hyperparameters

# G  Additional Results

## G.1  Impact of Replay Buffer Size on Learning Performance

The following figures show the impact of reducing replay buffer size of three state-of-the art deep RL algorithms — SAC, PPO and TD3; Reducing the size of the replay buffer has detrimental impact on learning performance. Each solid learning curve is an average of 30 independent runs. The shaded regions represent a 95% confidence interval. These learning curves were also used to generate Fig. 1.

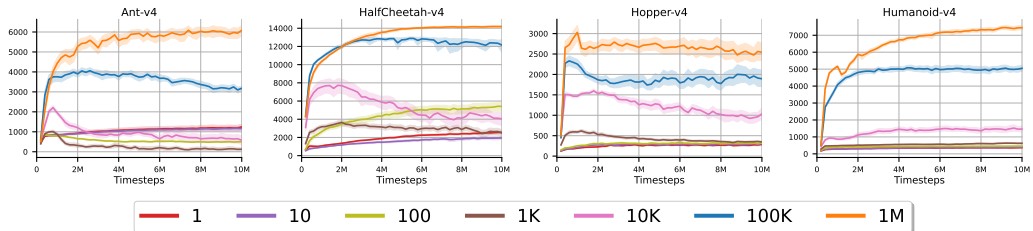

Figure 18: SAC

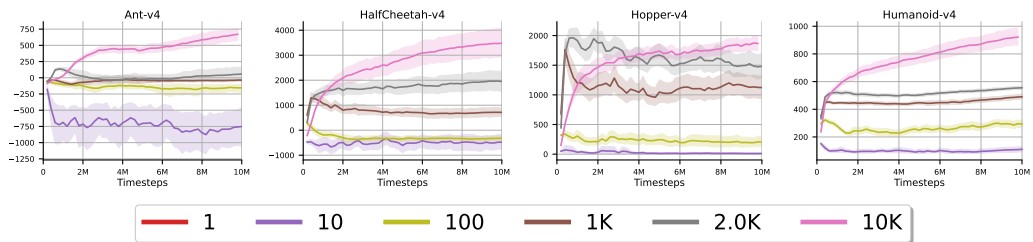

Figure 19: PPO

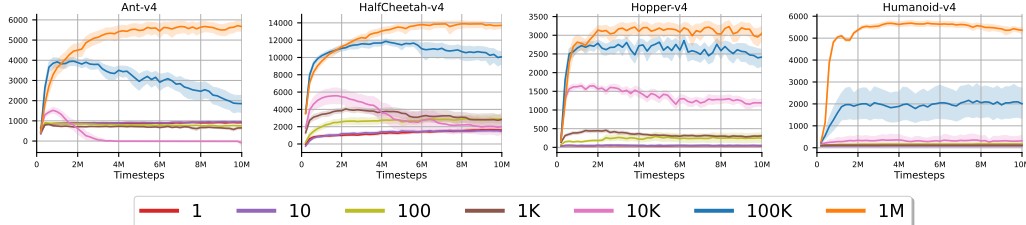

Figure 20: TD3

## H  Real-Robot Experiment Description

**UR-Reacher-2D**  We utilize the UR-Reacher-2 task, as developed by Mahmood et al. (2018), which involves the Reacher task using a UR5 robot. The agent aims to reach arbitrary target positions on a 2D plane. We control the second and third joints from the base by sending angular speeds within the range of $[-0.3, +0.3]rad/s$. The observation vector includes joint angles, joint velocities, the previous action, and the vector difference between the target and the fingertip coordinates. The workstation for UR5-VisualReacher has an AMD Ryzen Threadripper 2950 processor, an NVidia 2080Ti GPU, and 128G memory.

**Create-Mover**  We utilize the Create-Mover task, as developed by Mahmood et al. (2018), where the agent needs to move the robot forward as fast as possible within an enclosed arena. Compared to the original paper, we have a $3.92ft \times 4.33ft$ arena. The action space is $[-150mm/s, 150mm/s]^2$ for actuating the two wheels with speed control. The observation vector is composed of 6 wall-sensors values and the previous action. For the wall sensors, we always take the latest values received within the action cycle and use Equation 1 by (Benet et al. 2002) to convert the incoming signals to approximate distances. The reward function is the summation of the directed distance values over 10 most recent sensory packets. An episode is 90 seconds long but ends earlier if the agent triggers one of its bump sensors. When an episode terminates, the position of the robot is reset by moving backward to avoid bumping into the wall immediately.

| Parameter | AVG | SAC |
|---|---|---|
| Replay Buffer Size | 1 | 1 |
| Minibatch Size | 1 | 1 |
| Discount factor ($\gamma$) | 0.95 | 0.99 |
| Actor Learning Rate | $3 \times 10^{-4}$ | $3 \times 10^{-4}$ |
| Actor Learning Rate | 0.00087 | $3 \times 10^{-4}$ |
| Update Actor Every | 1 | 1 |
| Update Critic Every | 1 | 1 |
| Update Critic Target Every | N/A | 1 |
| Target Smoothing Coefficient ($\tau$) | N/A | 0.005 |
| Target Entropy | N/A | $-|\mathcal{A}|$ |
| Entropy coefficient ($\eta$) | 0.05 | Learnable parameter |
| Optimizer | Adam | Adam |

Table 6: Default parameters for Robot Tasks.

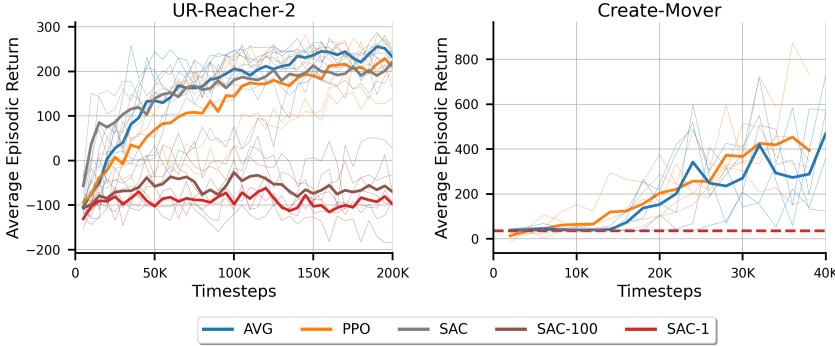

Figure 21: Learning curves on Robots. Comparison of AVG with full PPO & SAC. Note that running SAC and SAC-100 onboard for the Create-Mover task is computationally infeasible.

# I Convergence Analysis for Reparameterization Gradient

In this section, we present a convergence analysis for reparameterization policy gradient (RPG) in (2), which is one of the main components in our proposed AVG. We analyze a slightly different variant of AVG, that we call RPG-TD, shown in Algorithm 7. We extend the convergence result from Xiong et al. (2022) for deterministic policies to the general case of reparameterized policies.

Like AVG, RPG-TD uses the reparameterization gradient and updates one sample at a time, but it differs in that it does not have entropy regularization and normalizations. We also make a few typical theoretical assumptions, like i.i.d. sampling of transition tuples, that do not perfectly match the real setting for AVG. Following Xiong et al. (2022) and for analytical convenience, we use the stationary state distribution $d_\theta(s) = \lim_{T\to\infty} \int_{s_0} \frac{1}{T} \sum_{t=0}^{T} d_0(s_0) p(s_0 \to s, t, f_\theta) \, ds_0$ for the critic update, and the discounted state visitation $\nu_\theta(s) = \int_{s_0} \sum_{t=0}^{\infty} \gamma^t d_0(s_0) p(s_0 \to s, t, f_\theta) \, ds_0$ for the actor update. Here, $f_\theta(s, \epsilon)$ denotes the reparameterized policy, and $p(s_0 \to s, t, f_\theta)$ represents the density of state $s$ after $t$ steps from state $s_0$ following policy $f_\theta$. Note that we follow the notations and language from Xiong et al. (2022) while avoiding changes as much as possible for easy comparison with the original result.

$$\nabla J(\theta) = \int_s \int_\epsilon \nu_\theta(s) p(\epsilon) \nabla_\theta f_\theta(s, \epsilon) \nabla_a Q^{f_\theta}(s, a)|_{a=f_\theta(s,\epsilon)} \, d\epsilon ds$$
$$= \mathbb{E}_{\nu_\theta, p} \left[ \nabla_\theta f_\theta(s, \epsilon) \nabla_a Q^{f_\theta}(s, a)|_{a=f_\theta(s,\epsilon)} \right]. \tag{2}$$

We will present the assumptions and the convergence result for RPG-TD in Appendix I.1. The proofs of the convergence result and the intermediate results are provided in Appendix I.2. To highlight the differences between our extended analysis and that of Xiong et al. (2022), we use blue to indicate modifications specific to reparameterized policies. These modifications include replacing the deterministic policy $\mu_\theta(s)$ with the reparameterized policy $f_\theta(s, \epsilon)$ and properly handling of the expectation over the prior random variable $\epsilon \sim p$. In addition, we fixed a few errors in the original analysis and result, which are shown in red.

---

**Algorithm 7** RPG-TD

---

1: **Input:** $\alpha_w, \alpha_\theta, w_0, \theta_0$, batch size $M$.
2: **for** $t = 0, 1, \ldots, T$ **do**
3:     **for** $j = 0, 1, \ldots, M - 1$ **do**
4:         Sample $s_{t,j} \sim d_{\theta_t}, \epsilon_{t,j} \sim p$.
5:         Generate $a_{t,j} = f_{\theta_t}(s_{t,j}, \epsilon_{t,j})$.
6:         Sample $s_{t+1,j} \sim P(\cdot|s_{t,j}, a_{t,j}), \epsilon_{t+1,j} \sim p$, and $r_{t,j}$.
7:         Generate $a_{t+1,j} = f_{\theta_t}(s_{t+1,j}, \epsilon_{t+1,j})$.
8:         Denote $x_{t,j} = (s_{t,j}, a_{t,j})$.
9:         $\delta_{t,j} = r_{t,j} + \gamma \phi(x_{t+1,j})^T w_t - \phi(x_{t,j})^T w_t$.
10:     **end for**
11:     $w_{t+1} = w_t + \frac{\alpha_w}{M} \sum_{j=0}^{M-1} \delta_{t,j} \phi(x_{t,j})$.
12:     **for** $j = 0, 1, \ldots, M - 1$ **do**
13:         Sample $s'_{t,j} \sim \nu_{\theta_t}, \epsilon'_{t,j} \sim p$.
14:     **end for**
15:     $\theta_{t+1} = \theta_t + \frac{\alpha_\theta}{M} \sum_{j=0}^{M-1} \nabla_\theta f_{\theta_t}(s'_{t,j}, \epsilon'_{t,j}) \nabla_\theta f_{\theta_t}(s'_{t,j}, \epsilon'_{t,j})^T w_t$.
16: **end for**

---

## I.1 Convergence Result

We present the full set of assumptions below and refer interested reader to Xiong et al. (2022) for detailed discussions about these assumptions.

**Assumption 3.** For any $\theta_1, \theta_2, \theta \in \mathbb{R}^d$, there exist positive constants $L_f, L_\phi$ and $\lambda_\Phi$, such that (1) $\left\| f_{\theta_1}(s, \epsilon) - f_{\theta_2}(s, \epsilon) \right\| \leq L_f \left\| \theta_1 - \theta_2 \right\|, \forall s \in \mathcal{S}, \epsilon \in \mathbb{R}$; (2) $\left\| \nabla_\theta f_{\theta_1}(s, \epsilon) - \nabla_\theta f_{\theta_2}(s, \epsilon) \right\| \leq L_\psi \left\| \theta_1 - \theta_2 \right\|, \forall s \in \mathcal{S}, \epsilon \in \mathbb{R}$; (3) the matrix $\Psi_\theta := \mathbb{E}_{\nu_\theta, p} \left[ \nabla_\theta f_\theta(s, \epsilon) \nabla_\theta f_\theta(s, \epsilon)^T \right]$ is non-singular with the minimal eigenvalue uniformly lower-bounded as $\sigma_{\min}(\Psi_\theta) \geq \lambda_\Psi$.

**Assumption 4.** For any $a_1, a_2 \in \mathcal{A}$, there exist positive constants $L_P, L_r$, such that (1) the transition kernel satisfies $|P(s'|s, a_1) - P(s'|s, a_2)| \leq L_P \|a_1 - a_2\|, \forall s, s' \in \mathcal{S}$; (2) the reward function satisfies $|r(s, a_1) - r(s, a_2)| \leq L_r \|a_1 - a_2\|, \forall s, s' \in \mathcal{S}$.

**Assumption 5.** For any $a_1, a_2 \in \mathcal{A}$, there exists a positive constant $L_Q$, such that $\left\|\nabla_a Q^{f_\theta}(s, a_1) - \nabla_a Q^{f_\theta}(s, a_2)\right\| \leq L_Q \|a_1 - a_2\|, \forall \theta \in \mathbb{R}^d, s \in \mathcal{S}$.

**Assumption 6.** The feature function $\phi : \mathcal{S} \times \mathcal{A} \to \mathbb{R}^d$ is uniformly bounded, i.e., $\|\phi(\cdot, \cdot)\| \leq C_\phi$ for some positive constant $C_\phi$. In addition, we define $A = \mathbb{E}_{d_\theta}\left[\phi(x)(\gamma\phi(x') - \phi(x))^T\right]$ and $D = \mathbb{E}_{d_\theta}\left[\phi(x)\phi(x)^T\right]$, and assume that $A$ and $D$ are non-singular. We further assume that the absolute value of the eigenvalues of $A$ are uniformly lower bounded, i.e., $|\sigma(A)| \geq \lambda_A$ for some positive constant $\lambda_A$.

**Proposition 7** (Compatible function approximation). *A function estimator $Q^w(s, a)$ is compatible with a reparameterized policy $f_\theta$, i.e., $\nabla J(\theta) = \mathbb{E}_{\nu_\theta, p}\left[\nabla_\theta f_\theta(s, \epsilon)\nabla_a Q^w(s, a)|_{a = f_\theta(s, \epsilon)}\right]$, if it satisfies the following two conditions:*

1. $\nabla_a Q^w(s, a)|_{a = f_\theta(s, \epsilon)} = \nabla_\theta f_\theta(s, \epsilon)^T w$;

2. $w = w^*_{\xi_\theta}$ minimizes the mean square error $\mathbb{E}_{\nu_\theta, p}\left[\xi(s, \epsilon; \theta, w)^T \xi(s, \epsilon; \theta, w)\right]$, where $\xi(s, \epsilon; \theta, w) = \nabla_a Q^w(s, a)|_{a = f_\theta(s, \epsilon)} - \nabla_a Q^{f_\theta}(s, a)|_{a = f_\theta(s, \epsilon)}$.

Given the above assumption, one can show that the reparameterization gradient is smooth (Lemma 8), and that Algorithm 7 converges (Theorem 9), the proofs of which are presented in Appendix I.2.

**Lemma 8.** *Suppose Assumptions 3-5 hold. Then the reparameterization gradient $\nabla J(\theta)$ defined in (2) is Lipschitz continuous with the parameter $L_J$, i.e., $\forall \theta_1, \theta_2 \in \mathbb{R}^d$,*

$$\|\nabla J(\theta_1) - \nabla J(\theta_2)\| \leq L_J \|\theta_1 - \theta_2\|, \tag{3}$$

*where $L_J = \left(\frac{1}{2}L_P L_f^2 L_\nu C_\nu + \frac{L_\psi}{1-\gamma}\right)\left(L_r + \frac{\gamma R_{\max} L_P}{1-\gamma}\right) + \frac{L_f}{1-\gamma}\left(L_Q L_f + \frac{\gamma}{2}L_P^2 R_{\max} L_f C_\nu + \frac{\gamma L_P L_r L_f}{1-\gamma}\right)$.*

**Theorem 9.** *Suppose that Assumptions 3-6 hold. Let $\alpha_w \leq \frac{\lambda}{2C_A^2}; M \geq \frac{48\alpha_w C_A^2}{\lambda}; \alpha_\theta \leq$ $\min\left\{\frac{1}{4L_J}, \frac{\lambda\alpha_w}{24\sqrt{6}L_h L_w}\right\}$. Then the output of RPG-TD in Algorithm 7 satisfies*

$$\min_{t \in [T]} \mathbb{E}\|\nabla J(\theta_t)\|^2 \leq \frac{c_1}{T} + \frac{c_2}{M} + c_3 \kappa^2,$$

*where $c_1 = \frac{8R_{\max}}{\alpha_\theta(1-\gamma)} + \frac{144L_h^2}{\lambda\alpha_w}\left\|w_0 - w^*_{\theta_0}\right\|^2, c_2 = \left[48\alpha_w^2(C_A^2 C_w^2 + C_b^2) + \frac{96L_w^2 L_f^4 C_{w_\xi}^2 \alpha_\theta^2}{\lambda\alpha_w}\right] \cdot \frac{144L_h^2}{\lambda\alpha_w} +$ $72L_f^4 C_{w_\xi}^2, c_3 = 18L_h^2 + \left[\frac{24L_w^2 L_h^2 \alpha_\theta^2}{\lambda\alpha_w} + \frac{24}{\lambda\alpha_w}\right] \cdot \frac{144L_h^2}{\lambda\alpha_w}$ with $C_A = 2C_\phi^2, C_b = R_{\max} C_\phi, C_w = \frac{R_{\max} C_\phi}{\lambda_A}, C_{w_\xi} = \frac{L_f C_Q}{\lambda_\Psi(1-\gamma)}, L_w = \frac{L_J}{\lambda_\Psi} + \frac{L_f C_Q}{\lambda_\Psi^2(1-\gamma)}\left(L_f^2 L_\nu + \frac{2L_f L_\psi}{1-\gamma}\right), L_h = L_f^2, C_Q = L_r + L_P \cdot \frac{\gamma R_{\max}}{1-\gamma}, L_\nu = \frac{1}{2}C_\nu L_P L_f$, and $L_J$ defined in Lemma 8, and we define*

$$\kappa := \max_\theta \left\|w^*_\theta - w^*_{\xi_\theta}\right\|. \tag{4}$$

**Comparison with Theorem 9 and Theorem 1 of Xiong et al. (2022).** The differences between the reparameterization gradient and the deterministic policy gradient results are minimal. Aside from correcting the errors (highlighted in red; see Appendix I.2 for details), the most notable distinction is the replacement of $L_\mu$ in Xiong et al. (2022) with $L_f$, which may be different. Additionally, constants related to the critic, such as $L_Q$ and $\kappa$, may also differ, as they are now defined for a more general policy class. While this theoretical comparison shows little divergence, practical performance could vary significantly.

## I.2 Proofs

### I.2.1 Supporting Lemmas for Proving Lemma 8

**Lemma 10.** *Suppose Assumptions 3 and 4 hold. We define the total variation norm between two state visitation distributions respectively corresponding to two policies $f_{\theta_1}, f_{\theta_2}$ as $\|\nu_{\theta_1}(\cdot) - \nu_{\theta_2}(\cdot)\|_{TV} = \int_s |\nu_{\theta_1}(ds) - \nu_{\theta_2}(ds)|$. Then there exists some constant $L_\nu > 0$, such that*

$$\|\nu_{\theta_1}(\cdot) - \nu_{\theta_2}(\cdot)\|_{TV} \leq L_\nu \|\theta_1 - \theta_2\|.$$

*Proof.* Since we consider ergodic Markov chains, Theorem 3.1 of Mitrophanov (2005) shows that there exists some constant $C_\nu > 1$, such that

$$\left\| \nu_{\theta_1}(\cdot) - \nu_{\theta_2}(\cdot) \right\|_{TV} \leq C_\nu \left\| P_{\theta_1} - P_{\theta_2} \right\|_{\text{op}}, \tag{5}$$

where $P_\theta$ denotes the state transition kernel corresponding to a policy $f_\theta$, and the operator norm $\|\cdot\|_{\text{op}}$ is given by $\|P\|_{\text{op}} = \sup_{\|q\|_{TV}=1} \|qP\|_{TV}$. Then we have

$$
\begin{aligned}
\left\| P_{\theta_1} - P_{\theta_2} \right\|_{\text{op}} &= \sup_{\|q\|_{TV}=1} \left\| \int_s (P_{\theta_1} - P_{\theta_2})(s, \cdot) q(ds) \right\|_{TV} \\
&= \frac{1}{2} \sup_{\|q\|_{TV}=1} \int_{s'} \left| \int_s \left( P_{\theta_1}(s, ds') - P_{\theta_2}(s, ds') \right) q(ds) \right| \\
&\leq \frac{1}{2} \sup_{\|q\|_{TV}=1} \int_{s'} \int_s \left| P_{\theta_1}(s, ds') - P_{\theta_2}(s, ds') \right| q(ds) \\
&= \frac{1}{2} \sup_{\|q\|_{TV}=1} \int_{s'} \int_s \left| \int_\epsilon \left( P(ds'|s, f_{\theta_1}(s, \epsilon)) - P(ds'|s, f_{\theta_1}(s, \epsilon)) \right) p(d\epsilon) \right| q(ds) \\
&\leq \frac{1}{2} \sup_{\|q\|_{TV}=1} \int_{s'} \int_s \int_\epsilon \left| P(ds'|s, f_{\theta_1}(s, \epsilon)) - P(ds'|s, f_{\theta_1}(s, \epsilon)) \right| p(d\epsilon) q(ds) \\
&\overset{(i)}{\leq} \frac{1}{2} \sup_{\|q\|_{TV}=1} \int_s \int_\epsilon L_P \left\| f_{\theta_1}(s, \epsilon) - f_{\theta_2}(s, \epsilon) \right\| p(d\epsilon) q(ds) \\
&\overset{(ii)}{\leq} \frac{1}{2} L_P L_f \left\| \theta_1 - \theta_2 \right\|,
\end{aligned}
$$

where (i) follows form Assumption 4, and (ii) follows from Assumption 3. Then, combining the above bound together with (5) completes the proof. □

**Lemma 11.** *Suppose Assumptions 3 and 4 hold. The value function is Lipschitz continuous w.r.t. the policies. That is, for any $\theta_1, \theta_2 \in \mathbb{R}^d, s \in \mathcal{S}$, we have*

$$\left\| V^{f_{\theta_1}}(s) - V^{f_{\theta_2}}(s) \right\| \leq L_V \left\| \theta_1 - \theta_2 \right\|,$$

*where $L_V = R_{\max} L_\nu + \frac{L_r L_f}{1-\gamma}$.*

*Proof.* By definition, we have $V^{f_\theta}(s_0) = \int_s \int_\epsilon r(s, f_\theta(s, \epsilon)) p(d\epsilon) \nu_{f_\theta}^{s_0}(ds)$, where $\nu_{f_\theta}^{s_0}(\cdot)$ is the discounted state visitation measure given the initial state, i.e., $\nu_{f_\theta}^{s_0}(s) = \sum_{t=0}^\infty \gamma^t p(s_0 \to s, t, f_\theta)$. We then derive

$$
\begin{aligned}
&\left| V^{f_{\theta_1}}(s_0) - V^{f_{\theta_2}}(s_0) \right| \\
&= \left| \int_s \int_\epsilon r(s, f_{\theta_1}(s, \epsilon)) p(d\epsilon) \nu_{f_{\theta_1}}^{s_0}(ds) - \int_s \int_\epsilon r(s, f_{\theta_2}(s, \epsilon)) p(d\epsilon) \nu_{f_{\theta_2}}^{s_0}(ds) \right| \\
&\leq \int_\epsilon \left| \int_s r(s, f_{\theta_1}(s, \epsilon)) \nu_{f_{\theta_1}}^{s_0}(ds) - \int_s r(s, f_{\theta_2}(s, \epsilon)) \nu_{f_{\theta_2}}^{s_0}(ds) \right| p(d\epsilon) \\
&\leq \int_\epsilon \left| \int_s r(s, f_{\theta_1}(s, \epsilon)) \nu_{f_{\theta_1}}^{s_0}(ds) - \int_s r(s, f_{\theta_1}(s, \epsilon)) \nu_{f_{\theta_2}}^{s_0}(ds) \right| p(d\epsilon) \\
&\quad + \int_\epsilon \left| \int_s r(s, f_{\theta_1}(s, \epsilon)) \nu_{f_{\theta_2}}^{s_0}(ds) - \int_s r(s, f_{\theta_2}(s, \epsilon)) \nu_{f_{\theta_2}}^{s_0}(ds) \right| p(d\epsilon) \\
&\leq \int_\epsilon \int_s \left| r(s, f_{\theta_1}(s, \epsilon)) \right| \cdot \left| \nu_{f_{\theta_1}}^{s_0}(ds) - \nu_{f_{\theta_2}}^{s_0}(ds) \right| p(d\epsilon) \\
&\quad + \int_\epsilon \int_s \left| r(s, f_{\theta_1}(s, \epsilon)) - r(s, f_{\theta_2}(s, \epsilon)) \right| \nu_{f_{\theta_2}}^{s_0}(ds) p(d\epsilon) \\
&\overset{(i)}{\leq} R_{\max} \left\| \nu_{f_{\theta_1}}^{s_0}(\cdot) - \nu_{f_{\theta_2}}^{s_0}(\cdot) \right\|_{TV} + L_r \int_\epsilon \int_s \left\| f_{\theta_1}(s, \epsilon) - f_{\theta_2}(s, \epsilon) \right\| \nu_{f_{\theta_2}}^{s_0}(ds) p(d\epsilon)
\end{aligned}
$$

$$\overset{(ii)}{\leq} R_{\max} L_\nu \left\| \theta_1 - \theta_2 \right\| + L_r L_f \left\| \theta_1 - \theta_2 \right\| \int_\epsilon \int_s \nu^{s_0}_{f_{\theta_2}}(ds)p(d\epsilon)$$

$$= \left( R_{\max} L_\nu + \frac{L_r L_f}{1-\gamma} \right) \left\| \theta_1 - \theta_2 \right\|,$$

where (i) follows from Assumption 4, and (ii) follows from Lemma 10 and Assumption 3. □

**Lemma 12.** *Suppose Assumptions 3-5 hold. The gradient of Q-function w.r.t. action is uniformly bounded. That is, for any $(s,a) \in \mathcal{S} \times \mathcal{A}, \theta \in \mathbb{R}^d$,*

$$\left\| \nabla_a Q^{f_\theta}(s,a) \right\| \leq C_Q,$$

*where $C_Q = L_r + L_P \cdot \frac{\gamma R_{\max}}{1-\gamma}$. Furthermore, $\nabla_a Q^{f_\theta}(s,a_\theta)$ is Lipschitz continuous w.r.t. $\theta$, that is, for any $\theta_1, \theta_2 \in \mathbb{R}^d$, we have*

$$\left\| \nabla_a Q^{f_{\theta_1}}(s, f_{\theta_1}(s,\epsilon)) - \nabla_a Q^{f_{\theta_2}}(s, f_{\theta_2}(s,\epsilon)) \right\| \leq L'_Q \left\| \theta_1 - \theta_2 \right\|,$$

*where $L'_Q = L_Q L_f + \gamma L_P L_V$.*

*Proof.* For the boundedness property, we have

$$\left\| \nabla_a Q^{f_\theta}(s,a) \right\| = \left\| \nabla_a \int_s \left( r(s,a) + \gamma P(s'|s,a) V^{f_\theta}(s') \right) ds' \right\|$$

$$\leq \left\| \nabla_a r(s,a) \right\| + \gamma \int_s \left\| \nabla_a P(s'|s,a) \right\| \cdot \left| V^{f_\theta}(s') \right| ds'$$

$$\leq L_r + L_P \cdot \frac{\gamma R_{\max}}{1-\gamma},$$

where the last inequality follows from Assumptions 3, 4 and the fact that $\left| V^{f_\theta}(s') \right| \leq \frac{R_{\max}}{1-\gamma}$.

We next show the Lipschitz property as follows.

$$\left\| \nabla_a Q^{f_{\theta_1}}(s, f_{\theta_1}(s,\epsilon)) - \nabla_a Q^{f_{\theta_2}}(s, f_{\theta_2}(s,\epsilon)) \right\|$$

$$\leq \left\| \nabla_a Q^{f_{\theta_1}}(s, f_{\theta_1}(s,\epsilon)) - \nabla_a Q^{f_{\theta_1}}(s, f_{\theta_2}(s,\epsilon)) \right\|$$

$$+ \left\| \nabla_a Q^{f_{\theta_1}}(s, f_{\theta_2}(s,\epsilon)) - \nabla_a Q^{f_{\theta_2}}(s, f_{\theta_2}(s,\epsilon)) \right\|$$

$$\overset{(i)}{\leq} L_Q \left\| f_{\theta_1}(s,\epsilon) - f_{\theta_2}(s,\epsilon) \right\| + \left\| \nabla_a Q^{f_{\theta_1}}(s, f_{\theta_2}(s,\epsilon)) - \nabla_a Q^{f_{\theta_2}}(s, f_{\theta_2}(s,\epsilon)) \right\|$$

$$\overset{(ii)}{\leq} L_Q L_f \left\| \theta_1 - \theta_2 \right\| + \left\| \int_s \gamma \nabla_a P(s'|s,a) \left( V^{f_{\theta_1}}(s') - V^{f_{\theta_2}}(s') \right) ds' \right\|$$

$$\leq L_Q L_f \left\| \theta_1 - \theta_2 \right\| + \gamma \int_s \left\| \nabla_a P(s'|s,a) \right\| \cdot \left| V^{f_{\theta_1}}(s') - V^{f_{\theta_2}}(s') \right| ds'$$

$$\overset{(iii)}{\leq} \left( L_Q L_f + \gamma L_P L_V \right) \left\| \theta_1 - \theta_2 \right\|,$$

where (i) follows from Assumption 5, (ii) follows from Assumption 3 and (iii) follows from Assumption 4 and Lemma 11. □

### I.2.2 Proof of Lemma 8

To simplify the notation, we define $\psi_\theta(s,\epsilon) := \nabla_\theta f_\theta(s,\epsilon)$, $a_\theta = f_\theta(s,\epsilon)$ and $\nabla_a Q^{f_\theta}(s,a_\theta) = \nabla_a Q^{f_\theta}(s,a)|_{a=f_\theta(s,\epsilon)}$ in the following proof.

We start from the form of the off-policy deterministic policy gradient given in (2), and have

$$\left\| \nabla J(\theta_1) - \nabla J(\theta_2) \right\|$$

$$= \left\| \int_s \int_\epsilon \psi_{\theta_1}(s,\epsilon) \nabla_a Q^{f_{\theta_1}}(s, a_{\theta_1}) p(d\epsilon) \nu_{\theta_1}(ds) - \int_s \int_\epsilon \psi_{\theta_2}(s,\epsilon) \nabla_a Q^{f_{\theta_2}}(s, a_{\theta_2}) p(d\epsilon) \nu_{\theta_2}(ds) \right\|$$

$$\leq \int_\epsilon \left\| \int_s \psi_{\theta_1}(s,\epsilon)\nabla_a Q^{f_{\theta_1}}(s,a_{\theta_1})\nu_{\theta_1}(ds) - \int_s \psi_{\theta_2}(s,\epsilon)\nabla_a Q^{f_{\theta_2}}(s,a_{\theta_2})\nu_{\theta_2}(ds) \right\| p(d\epsilon). \tag{6}$$

Now,

$$\left\| \int_s \psi_{\theta_1}(s,\epsilon)\nabla_a Q^{f_{\theta_1}}(s,a_{\theta_1})\nu_{\theta_1}(ds) - \int_s \psi_{\theta_2}(s,\epsilon)\nabla_a Q^{f_{\theta_2}}(s,a_{\theta_2})\nu_{\theta_2}(ds) \right\|$$

$$= \left\| \int_s \psi_{\theta_1}(s,\epsilon)\nabla_a Q^{f_{\theta_1}}(s,a_{\theta_1})\nu_{\theta_1}(ds) - \int_s \psi_{\theta_1}(s,\epsilon)\nabla_a Q^{f_{\theta_1}}(s,a_{\theta_1})\nu_{\theta_2}(ds) \right.$$

$$+ \int_s \psi_{\theta_1}(s,\epsilon)\nabla_a Q^{f_{\theta_1}}(s,a_{\theta_1})\nu_{\theta_2}(ds) - \int_s \psi_{\theta_1}(s,\epsilon)\nabla_a Q^{f_{\theta_2}}(s,a_{\theta_2})\nu_{\theta_2}(ds)$$

$$\left. + \int_s \psi_{\theta_1}(s,\epsilon)\nabla_a Q^{f_{\theta_2}}(s,a_{\theta_2})\nu_{\theta_2}(ds) - \int_s \psi_{\theta_2}(s,\epsilon)\nabla_a Q^{f_{\theta_2}}(s,a_{\theta_2})\nu_{\theta_2}(ds) \right\|$$

$$\leq \left\| \int_s \psi_{\theta_1}(s,\epsilon)\nabla_a Q^{f_{\theta_1}}(s,a_{\theta_1})\nu_{\theta_1}(ds) - \int_s \psi_{\theta_1}(s,\epsilon)\nabla_a Q^{f_{\theta_1}}(s,a_{\theta_1})\nu_{\theta_2}(ds) \right\|$$

$$+ \left\| \int_s \psi_{\theta_1}(s,\epsilon)\nabla_a Q^{f_{\theta_1}}(s,a_{\theta_1})\nu_{\theta_2}(ds) - \int_s \psi_{\theta_1}(s,\epsilon)\nabla_a Q^{f_{\theta_2}}(s,a_{\theta_2})\nu_{\theta_2}(ds) \right\|$$

$$+ \left\| \int_s \psi_{\theta_1}(s,\epsilon)\nabla_a Q^{f_{\theta_2}}(s,a_{\theta_2})\nu_{\theta_2}(ds) - \int_s \psi_{\theta_2}(s,\epsilon)\nabla_a Q^{f_{\theta_2}}(s,a_{\theta_2})\nu_{\theta_2}(ds) \right\|$$

$$\leq \int_s \|\psi_{\theta_1}(s,\epsilon)\| \cdot \left\| \nabla_a Q^{f_{\theta_1}}(s,a_{\theta_1}) \right\| |\nu_{\theta_1}(ds) - \nu_{\theta_2}(ds)|$$

$$+ \int_s \|\psi_{\theta_1}(s,\epsilon)\| \cdot \left\| \nabla_a Q^{f_{\theta_1}}(s,a_{\theta_1}) - \nabla_a Q^{f_{\theta_2}}(s,a_{\theta_2}) \right\| \nu_{\theta_2}(ds)$$

$$+ \int_s \|\psi_{\theta_1}(s,\epsilon) - \psi_{\theta_2}(s,\epsilon)\| \cdot \left\| \nabla_a Q^{f_{\theta_2}}(s,a_{\theta_2}) \right\| \nu_{\theta_2}(ds)$$

$$\overset{(i)}{\leq} L_f C_Q \|\nu_{\theta_1}(\cdot) - \nu_{\theta_2}(\cdot)\|_{TV} + L_f \int_s \left\| \nabla_a Q^{f_{\theta_1}}(s,a_{\theta_1}) - \nabla_a Q^{f_{\theta_2}}(s,a_{\theta_2}) \right\| \nu_{\theta_2}(ds)$$

$$+ C_Q \int_s \|\psi_{\theta_1}(s,\epsilon) - \psi_{\theta_2}(s,\epsilon)\| \nu_{\theta_2}(ds)$$

$$\overset{(ii)}{\leq} L_f C_Q \|\nu_{\theta_1}(\cdot) - \nu_{\theta_2}(\cdot)\|_{TV} + L_f L'_Q \|\theta_1 - \theta_2\| \int_s \nu_{\theta_2}(ds) + C_Q L_\psi \|\theta_1 - \theta_2\| \int_s \nu_{\theta_2}(ds)$$

$$\overset{(iii)}{=} L_f C_Q \|\nu_{\theta_1}(\cdot) - \nu_{\theta_2}(\cdot)\|_{TV} + \frac{L_f L'_Q}{1-\gamma} \|\theta_1 - \theta_2\| + \frac{C_Q L_\psi}{1-\gamma} \|\theta_1 - \theta_2\|$$

$$\overset{(iv)}{\leq} \left( L_f C_Q L_\nu + \frac{L_f L'_Q}{1-\gamma} + \frac{C_Q L_\psi}{1-\gamma} \right) \|\theta_1 - \theta_2\|$$

$$:= L_J \|\theta_1 - \theta_2\|, \tag{7}$$

where (i) follows because $\|\psi_\theta(s,\epsilon)\| \leq L_f$ as indicated by Assumption 3 and $\left\| \nabla_a Q^{f_\theta}(s,a) \right\| \leq C_Q$ by Lemma 12, (ii) follows from Assumption 3 and Lemma 12, (iii) follows because $\int_s \nu_\theta(ds) = \frac{1}{1-\gamma}$, and (iv) follows from Lemma 10. Plugging (7) to (6), we finish the proof.

### I.2.3  Supporting Lemmas for Proving Theorem 9

**Lemma 13.** *The following two properties hold.*

1. *Let $\hat{Y}, \bar{Y} \in \mathbb{R}^{d_1 \times d_2}$ be matrices satisfying $\left\| \hat{Y} \right\|_F \leq C_Y, \left\| \bar{Y} \right\|_F \leq C_Y$. If $\hat{Y}$ is an unbiased estimator of $\bar{Y}$ and $\{\hat{Y}_j\}_j$ are i.i.d. estimators, then we have*

$$\mathbb{E} \left\| \frac{1}{M} \sum_{j=0}^{M-1} \hat{Y}_j - \bar{Y} \right\|_F^2 \leq \frac{4C_Y^2}{M}.$$

2. Let $\hat{y}, \bar{y} \in \mathbb{R}^d$ be vectors satisfying $\|\hat{y}\| \leq C_y, \|\bar{y}\| \leq C_y$. If $\hat{y}$ is an unbiased estimator of $\bar{y}$ and $\{y_j\}_j$ are i.i.d. estimators, then we have

$$\mathbb{E} \left\| \frac{1}{M} \sum_{j=0}^{M-1} \hat{y}_j - \bar{y} \right\|^2 \leq \frac{4C_y^2}{M}.$$

*Proof.* See the proof of Lemma 4 of Xiong et al. (2022).

$\square$

**Lemma 14.** *Let $w^*_{\xi_\theta}$ be defined in Proposition 7. Suppose Assumptions 3-5 hold. Then we have*

$$\left\| w^*_{\xi_\theta} \right\| \leq C_{w_\xi},$$

*where $C_{w_\xi} = \frac{L_f C_Q}{\lambda_\Psi (1-\gamma)}$. Furthermore, for any $\theta_1, \theta_2$, we have*

$$\left\| w^*_{\xi_{\theta_1}} - w^*_{\xi_{\theta_2}} \right\| \leq L_w \|\theta_1 - \theta_2\|,$$

*where $L_w = \frac{L_J}{\lambda_\Psi} + \frac{L_f C_Q}{\lambda_\Psi^2 (1-\gamma)} \left( L_f^2 L_\nu + \frac{2L_f L_\psi}{1-\gamma} \right)$.*

*Proof.* We first show the boundedness of $\|\nabla J(\theta)\|$.

$$
\begin{aligned}
\|\nabla J(\theta)\| &= \left\| \int_s \int_\epsilon \nabla_\theta f_\theta(s,\epsilon) \nabla_a Q^{f_\theta}(s,a)|_{a=f_\theta(s,\epsilon)} p(d\epsilon) \nu_\theta(ds) \right\| \\
&\leq \int_s \int_\epsilon \|\nabla_\theta f_\theta(s,\epsilon)\| \left\| \nabla_a Q^{f_\theta}(s,a)|_{a=f_\theta(s,\epsilon)} \right\| p(d\epsilon) \nu_\theta(ds) \\
&\overset{(i)}{\leq} L_f C_Q \int_s \int_\epsilon \nu_\theta(ds) p(d\epsilon) = \frac{L_f C_Q}{(1-\gamma)},
\end{aligned}
\tag{8}
$$

where (i) follows from Assumption 3 and Lemma 12.

Recall we define $\Psi_\theta = \mathbb{E}_{\nu_{f_\theta}} \left[ \nabla_\theta f_\theta(s,\epsilon) \nabla_\theta f_\theta(s,\epsilon)^T \right]$. Assumption 3 implies that $\Psi_\theta$ is non-singular. Then by definition, we have

$$\left\| w^*_{\xi_\theta} \right\| = \left\| \Psi_\theta^{-1} \nabla J(\theta) \right\| \leq \frac{1}{\lambda_\Psi} \|\nabla J(\theta)\| \leq \frac{L_f C_Q}{\lambda_\Psi (1-\gamma)}.$$

Next, we show the Lipschitz continuity property.

$$
\begin{aligned}
&\left\| w^*_{\xi_{\theta_1}} - w^*_{\xi_{\theta_2}} \right\| \\
&= \left\| \Psi_{\theta_1}^{-1} \nabla J(\theta_1) - \Psi_{\theta_2}^{-1} \nabla J(\theta_2) \right\| \\
&= \left\| \Psi_{\theta_1}^{-1} \nabla J(\theta_1) - \Psi_{\theta_1}^{-1} \nabla J(\theta_2) + \Psi_{\theta_1}^{-1} \nabla J(\theta_2) - \Psi_{\theta_2}^{-1} \nabla J(\theta_2) \right\| \\
&\leq \left\| \Psi_{\theta_1}^{-1} (\nabla J(\theta_1) - \nabla J(\theta_2)) \right\| + \left\| \left( \Psi_{\theta_1}^{-1} - \Psi_{\theta_2}^{-1} \right) \nabla J(\theta_2) \right\| \\
&\overset{(i)}{\leq} \frac{L_J}{\lambda_\Psi} \|\theta_1 - \theta_2\| + \left\| \left( \Psi_{\theta_1}^{-1} - \Psi_{\theta_2}^{-1} \right) \nabla J(\theta_2) \right\| \\
&= \frac{L_J}{\lambda_\Psi} \|\theta_1 - \theta_2\| + \left\| \left( \Psi_{\theta_1}^{-1} \Psi_{\theta_2} \Psi_{\theta_2}^{-1} - \Psi_{\theta_1}^{-1} \Psi_{\theta_1} \Psi_{\theta_2}^{-1} \right) \nabla J(\theta_2) \right\| \\
&= \frac{L_J}{\lambda_\Psi} \|\theta_1 - \theta_2\| + \left\| \Psi_{\theta_1}^{-1} (\Psi_{\theta_2} - \Psi_{\theta_1}) \Psi_{\theta_2}^{-1} \nabla J(\theta_2) \right\| \\
&\leq \frac{L_J}{\lambda_\Psi} \|\theta_1 - \theta_2\| + \frac{1}{\lambda_\Psi^2} \|\Psi_{\theta_2} - \Psi_{\theta_1}\| \|\nabla J(\theta_2)\| \\
&\leq \frac{L_J}{\lambda_\Psi} \|\theta_1 - \theta_2\| + \frac{L_f C_Q}{\lambda_\Psi^2 (1-\gamma)} \|\Psi_{\theta_2} - \Psi_{\theta_1}\|,
\end{aligned}
$$

where (i) follows from Lemma 8 and Assumption 3.

Observe that

$$\|\Psi_{\theta_2} - \Psi_{\theta_1}\|$$

$$= \left\| \int_s \int_\epsilon \nabla_\theta f_{\theta_2}(s,\epsilon) \nabla_\theta f_{\theta_2}(s,\epsilon)^T p(d\epsilon) \nu_{\theta_2}(ds) - \int_s \int_\epsilon \nabla_\theta f_{\theta_1}(s,\epsilon) \nabla_\theta f_{\theta_1}(s,\epsilon)^T p(d\epsilon) \nu_{\theta_1}(ds) \right\|$$

$$= \left\| \int_\epsilon \left( \int_s \nabla_\theta f_{\theta_2}(s,\epsilon) \nabla_\theta f_{\theta_2}(s,\epsilon)^T \nu_{\theta_2}(ds) - \int_s \nabla_\theta f_{\theta_1}(s,\epsilon) \nabla_\theta f_{\theta_1}(s,\epsilon)^T \nu_{\theta_1}(ds) \right) p(d\epsilon) \right\|$$

$$\le \int_\epsilon \left\| \int_s \nabla_\theta f_{\theta_2}(s,\epsilon) \nabla_\theta f_{\theta_2}(s,\epsilon)^T \nu_{\theta_2}(ds) - \int_s \nabla_\theta f_{\theta_1}(s,\epsilon) \nabla_\theta f_{\theta_1}(s,\epsilon)^T \nu_{\theta_1}(ds) \right\| p(d\epsilon). \qquad (9)$$

Now,

$$\left\| \int_s \nabla_\theta f_{\theta_2}(s,\epsilon) \nabla_\theta f_{\theta_2}(s,\epsilon)^T \nu_{\theta_2}(ds) - \int_s \nabla_\theta f_{\theta_1}(s,\epsilon) \nabla_\theta f_{\theta_1}(s,\epsilon)^T \nu_{\theta_1}(ds) \right\|$$

$$\le \left\| \int_s \nabla_\theta f_{\theta_2}(s,\epsilon) \nabla_\theta f_{\theta_2}(s,\epsilon)^T \nu_{\theta_2}(ds) - \int_s \nabla_\theta f_{\theta_2}(s,\epsilon) \nabla_\theta f_{\theta_2}(s,\epsilon)^T \nu_{\theta_1}(ds) \right\|$$

$$+ \left\| \int_s \nabla_\theta f_{\theta_2}(s,\epsilon) \nabla_\theta f_{\theta_2}(s,\epsilon)^T \nu_{\theta_1}(ds) - \int_s \nabla_\theta f_{\theta_2}(s,\epsilon) \nabla_\theta f_{\theta_1}(s,\epsilon)^T \nu_{\theta_1}(ds) \right\|$$

$$+ \left\| \int_s \nabla_\theta f_{\theta_2}(s,\epsilon) \nabla_\theta f_{\theta_1}(s,\epsilon)^T \nu_{\theta_1}(ds) - \int_s \nabla_\theta f_{\theta_1}(s,\epsilon) \nabla_\theta f_{\theta_1}(s,\epsilon)^T \nu_{\theta_1}(ds) \right\|$$

$$\overset{(i)}{\le} L_f^2 \|\nu_{\theta_1}(\cdot) - \nu_{\theta_2}(\cdot)\|_{TV} + 2L_f \int_s \left\| \nabla_\theta f_{\theta_2}(s,\epsilon) - \nabla_\theta f_{\theta_1}(s,\epsilon) \right\| \nu_{\theta_1}(ds)$$

$$\overset{(ii)}{\le} L_f^2 \|\nu_{\theta_1}(\cdot) - \nu_{\theta_2}(\cdot)\|_{TV} + \frac{2L_f L_\psi}{1-\gamma} \|\theta_1 - \theta_2\|$$

$$\overset{(iii)}{\le} \left( L_f^2 L_\nu + \frac{2L_f L_\psi}{1-\gamma} \right) \|\theta_1 - \theta_2\|, \qquad (10)$$

where both (i) and (ii) follow from Assumption 3, and (iii) follows from Lemma 10. Plugging (9) to (10), we get

$$\|\Psi_{\theta_2} - \Psi_{\theta_1}\| \le \left( L_f^2 L_\nu + \frac{2L_f L_\psi}{1-\gamma} \right) \|\theta_1 - \theta_2\|.$$

Thus, we have

$$\left\| w_{\xi_{\theta_1}}^* - w_{\xi_{\theta_2}}^* \right\|$$

$$\le \frac{L_J}{\lambda_\Psi} \|\theta_1 - \theta_2\| + \frac{L_f C_Q}{\lambda_\Psi^2 (1-\gamma)} \|\Psi_{\theta_2} - \Psi_{\theta_1}\|$$

$$\le \left[ \frac{L_J}{\lambda_\Psi} + \frac{L_f C_Q}{\lambda_\Psi^2 (1-\gamma)} \left( L_f^2 L_\nu + \frac{2L_f L_\psi}{1-\gamma} \right) \right] \|\theta_1 - \theta_2\|.$$

$\square$

For the clarity of the presentation, we will use the following notation for the gradient estimate for $J(\theta_t)$:

$$h_{\theta_t}(w_t, \mathcal{B}_t) = \frac{1}{M} \sum_{j=0}^{M-1} \nabla_\theta f_{\theta_t}(s_{t,j}', \epsilon_{t,j}') \nabla_\theta f_{\theta_t}(s_{t,j}', \epsilon_{t,j}')^T w_t.$$

**Lemma 15.** *Suppose Assumptions 3-5. Then we have*

$$\mathbb{E} \|h_{\theta_t}(w_t, \mathcal{B}_t) - \nabla J(\theta_t)\|^2 \le 3L_h^2 \mathbb{E} \left\| w_t - w_{\theta_t}^* \right\|^2 + 3L_h^2 \kappa^2 + \frac{12 L_f^4 C_{w_\xi}^2}{M},$$

*where $L_h = L_f^2$ and $\kappa$ is defined in (4).*

*Proof.* By definition, we have

$$\mathbb{E}\left\|h_{\theta_t}(w_t, \mathcal{B}_t) - \nabla J(\theta_t)\right\|^2$$

$$= \mathbb{E}\left\|h_{\theta_t}(w_t, \mathcal{B}_t) - h_{\theta_t}(w_{\theta_t}^*, \mathcal{B}_t) + h_{\theta_t}(w_{\theta_t}^*, \mathcal{B}_t) - h_{\theta_t}(w_{\xi_{\theta_t}}^*, \mathcal{B}_t) + h_{\theta_t}(w_{\xi_{\theta_t}}^*, \mathcal{B}_t) - \nabla J(\theta_t)\right\|^2$$

$$\leq 3\mathbb{E}\left\|h_{\theta_t}(w_t, \mathcal{B}_t) - h_{\theta_t}(w_{\theta_t}^*, \mathcal{B}_t)\right\|^2 + 3\mathbb{E}\left\|h_{\theta_t}(w_{\theta_t}^*, \mathcal{B}_t) - h_{\theta_t}(w_{\xi_{\theta_t}}^*, \mathcal{B}_t)\right\|^2$$

$$+ 3\mathbb{E}\left\|h_{\theta_t}(w_{\xi_{\theta_t}}^*, \mathcal{B}_t) - \nabla J(\theta_t)\right\|^2$$

$$\overset{(i)}{\leq} 3L_h^2\mathbb{E}\left\|w_t - w_{\theta_t}^*\right\|^2 + 3L_h^2\mathbb{E}\left\|w_{\theta_t}^* - w_{\xi_{\theta_t}}^*\right\|^2 + 3\mathbb{E}\left\|h_{\theta_t}(w_{\xi_{\theta_t}}^*, \mathcal{B}_t) - \nabla J(\theta_t)\right\|^2$$

$$\overset{(ii)}{\leq} 3L_h^2\mathbb{E}\left\|w_t - w_{\theta_t}^*\right\|^2 + 3L_h^2\kappa^2 + 3\mathbb{E}\left\|h_{\theta_t}(w_{\xi_{\theta_t}}^*, \mathcal{B}_t) - \nabla J(\theta_t)\right\|^2$$

$$\overset{(iii)}{\leq} 3L_h^2\mathbb{E}\left\|w_t - w_{\theta_t}^*\right\|^2 + 3L_h^2\kappa^2 + \frac{12L_f^4 C_{w_\xi}^2}{M},$$

where (i) follows because for any $w_1, w_2, \theta \in \mathbb{R}^d$, we have

$$\left\|h_\theta(w_1, \mathcal{B}_t) - h_\theta(w_2, \mathcal{B}_t)\right\| = \left\|\frac{1}{M}\sum_{j=0}^{M-1}\nabla_\theta f_{\theta_t}(s'_{t,j}, \epsilon'_{t,j})\nabla_\theta f_{\theta_t}(s'_{t,j}, \epsilon'_{t,j})^T(w_1 - w_2)\right\|$$

$$\leq L_f^2\left\|w_1 - w_2\right\| := L_h\left\|w_1 - w_2\right\|,$$

(ii) follows from (4), and (iii) holds due to the fact that

$$\mathbb{E}\left\|h_{\theta_t}(w_{\xi_{\theta_t}}^*, \mathcal{B}_t) - \nabla J(\theta_t)\right\|^2$$

$$= \mathbb{E}\left\|\frac{1}{M}\sum_{j=0}^{M-1}\nabla_\theta f_{\theta_t}(s'_{t,j}, \epsilon'_{t,j})\nabla_\theta f_{\theta_t}(s'_{t,j}, \epsilon'_{t,j})^T w_{\xi_{\theta_t}}^* - \nabla J(\theta_t)\right\|^2$$

$$= \frac{1}{M^2}\sum_{i=0}^{M-1}\sum_{j=0}^{M-1}\mathbb{E}\langle\nabla_\theta f_{\theta_t}(s'_{t,i}, \epsilon'_{t,i})\nabla_\theta f_{\theta_t}(s'_{t,i}, \epsilon'_{t,i})^T w_{\xi_{\theta_t}}^* - \nabla J(\theta_t),$$

$$\nabla_\theta f_{\theta_t}(s'_{t,j}, \epsilon'_{t,j})\nabla_\theta f_{\theta_t}(s'_{t,j}, \epsilon'_{t,j})^T w_{\xi_{\theta_t}}^* - \nabla J(\theta_t)\rangle$$

$$= \frac{1}{M^2}\sum_{j=0}^{M-1}\mathbb{E}\left\|\nabla_\theta f_{\theta_t}(s'_{t,j}, \epsilon'_{t,j})\nabla_\theta f_{\theta_t}(s'_{t,j}, \epsilon'_{t,j})^T w_{\xi_{\theta_t}}^* - \nabla J(\theta_t)\right\|^2$$

$$\overset{(i)}{\leq} \frac{1}{M^2}\sum_{j=0}^{M-1}4L_f^4 C_{w_\xi}^2 = \frac{4L_f^4 C_{w_\xi}^2}{M},$$

where (i) follows from Assumption 3, Lemma 13 and Lemma 14.[4] Here, to apply Lemma 13, we need to upper-bound the norms of both the unbiased estimators and their expectation. For the former, we have

$$\left\|\nabla_\theta f_{\theta_t}(s'_{t,j}, \epsilon'_{t,j})\nabla_\theta f_{\theta_t}(s'_{t,j}, \epsilon'_{t,j})^T w_{\xi_{\theta_t}}^*\right\| \leq L_f^2 C_{w_\xi},$$

while for the latter, we have

$$\|\nabla J(\theta_t)\| \leq \frac{L_f C_Q}{(1-\gamma)} = \lambda_\Psi C_{w_\xi} \leq L_f^2 C_{w_\xi},$$

where we use the bound of $\nabla J(\theta_t)$ derived in (8), the definition of $C_{w_\xi}$ in Lemma 14, and that

$$\lambda_\Psi \leq \frac{1}{n}\text{trace}(\Psi) = \left\|\mathbb{E}_{\nu_\theta, p}\left[\nabla_\theta f_{\theta_t}(s, \epsilon)\nabla_\theta f_{\theta_t}(s, \epsilon)^T\right]\right\|^2 \leq L_f^2.$$

---

[4]In Xiong et al. (2022), the constant of the last equation is 2, without derivation. Here, we correct this constant to 4 following our detailed derivation.

$$\mathbb{E}\left\|\nabla_\theta f_{\theta_t}(s'_{t,j}, \epsilon'_{t,j})\nabla_\theta f_{\theta_t}(s'_{t,j}, \epsilon'_{t,j})^T w^*_{\xi_{\theta_t}} - \nabla J(\theta_t)\right\|^2 \leq 4L_f^4 C_{w_\epsilon}^2.$$

$\square$

### I.2.4 Proof of Theorem 9

We use the following notations for the clarity of the presentation:

$$g_{\theta_t}(w_t, \mathcal{B}_t) = \frac{1}{M}\sum_{j=0}^{M-1}\delta_{t,j}\phi(x_{t,j}) = \frac{1}{M}\sum_{j=0}^{M-1}(A_{t,j}w_t + b_{t,j}) := \hat{A}_t w_t + \hat{b}_t;$$

$$\bar{g}_{\theta_t}(w_t) = \mathbb{E}_{d_{\theta_t}}[\delta_t \phi(x_t)] = \bar{A}_t w_t + \bar{b}_t;$$

$$\bar{g}_{\theta_t}(w^*_{\theta_t}) = \bar{A}_t w^*_{\theta_t} + \bar{b}_t = 0.$$

**Step I: Characterizing dynamics of critic's error via coupling with actor.**

In the first step, we characterize the propagation of the dynamics of critic's dynamic tracking error based on its coupling with actor's updates. That is, we develop the relationship between $\left\|w_{t+1} - w^*_{\theta_{t+1}}\right\|^2$ and $\left\|w_t - w^*_{\theta_t}\right\|^2$ by their coupling with actor's updates.

We first use the dynamics of the critic to obtain

$$\begin{aligned}
&\left\|w_{t+1} - w^*_{\theta_t}\right\|^2 \\
&= \left\|w_t + \alpha_w g_{\theta_t}(w_t, \mathcal{B}_t) - w^*_{\theta_t}\right\|^2 \\
&= \left\|w_t - w^*_{\theta_t}\right\|^2 + 2\alpha_w\langle w_t - w^*_{\theta_t}, g_{\theta_t}(w_t, \mathcal{B}_t)\rangle + \alpha_w^2\left\|g_{\theta_t}(w_t, \mathcal{B}_t)\right\|^2 \\
&= \left\|w_t - w^*_{\theta_t}\right\|^2 + 2\alpha_w\langle w_t - w^*_{\theta_t}, \bar{g}_{\theta_t}(w_t)\rangle + 2\alpha_w\langle w_t - w^*_{\theta_t}, g_{\theta_t}(w_t, \mathcal{B}_t) - \bar{g}_{\theta_t}(w_t)\rangle \\
&\quad + \alpha_w^2\left\|g_{\theta_t}(w_t, \mathcal{B}_t)\right\|^2 \\
&= \left\|w_t - w^*_{\theta_t}\right\|^2 + 2\alpha_w(w_t - w^*_{\theta_t})^T\bar{A}_t(w_t - w^*_{\theta_t}) + 2\alpha_w\langle w_t - w^*_{\theta_t}, g_{\theta_t}(w_t, \mathcal{B}_t) - \bar{g}_{\theta_t}(w_t)\rangle \\
&\quad + \alpha_w^2\left\|g_{\theta_t}(w_t, \mathcal{B}_t)\right\|^2 \\
&\overset{(i)}{\leq} (1 - 2\alpha_w\lambda)\left\|w_t - w^*_{\theta_t}\right\|^2 + 2\alpha_w\langle w_t - w^*_{\theta_t}, g_{\theta_t}(w_t, \mathcal{B}_t) - \bar{g}_{\theta_t}(w_t)\rangle + \alpha_w^2\left\|g_{\theta_t}(w_t, \mathcal{B}_t)\right\|^2 \\
&\leq (1 - 2\alpha_w\lambda)\left\|w_t - w^*_{\theta_t}\right\|^2 + 2\alpha_w\langle w_t - w^*_{\theta_t}, g_{\theta_t}(w_t, \mathcal{B}_t) - \bar{g}_{\theta_t}(w_t)\rangle \\
&\quad + 2\alpha_w^2\left\|g_{\theta_t}(w_t, \mathcal{B}_t) - \bar{g}_{\theta_t}(w_t)\right\|^2 + 2\alpha_w^2\left\|\bar{g}_{\theta_t}(w_t)\right\|^2 \\
&= (1 - 2\alpha_w\lambda)\left\|w_t - w^*_{\theta_t}\right\|^2 + 2\alpha_w\langle w_t - w^*_{\theta_t}, g_{\theta_t}(w_t, \mathcal{B}_t) - \bar{g}_{\theta_t}(w_t)\rangle \\
&\quad + 2\alpha_w^2\left\|g_{\theta_t}(w_t, \mathcal{B}_t) - \bar{g}_{\theta_t}(w_t)\right\|^2 + 2\alpha_w^2\left\|\bar{g}_{\theta_t}(w_t) - \bar{g}_{\theta_t}(w^*_{\theta_t})\right\|^2 \\
&= (1 - 2\alpha_w\lambda)\left\|w_t - w^*_{\theta_t}\right\|^2 + 2\alpha_w\langle w_t - w^*_{\theta_t}, g_{\theta_t}(w_t, \mathcal{B}_t) - \bar{g}_{\theta_t}(w_t)\rangle \\
&\quad + 2\alpha_w^2\left\|g_{\theta_t}(w_t, \mathcal{B}_t) - \bar{g}_{\theta_t}(w_t)\right\|^2 + 2\alpha_w^2\left\|\bar{A}_t(w_t - w^*_{\theta_t})\right\|^2 \\
&\leq (1 - 2\alpha_w\lambda)\left\|w_t - w^*_{\theta_t}\right\|^2 + 2\alpha_w\langle w_t - w^*_{\theta_t}, g_{\theta_t}(w_t, \mathcal{B}_t) - \bar{g}_{\theta_t}(w_t)\rangle \\
&\quad + 2\alpha_w^2\left\|g_{\theta_t}(w_t, \mathcal{B}_t) - \bar{g}_{\theta_t}(w_t)\right\|^2 + 2\alpha_w^2\left\|\bar{A}_t\right\|^2\left\|(w_t - w^*_{\theta_t})\right\|^2 \\
&\overset{(ii)}{\leq} (1 - 2\alpha_w\lambda + 2\alpha_w^2 C_A^2)\left\|w_t - w^*_{\theta_t}\right\|^2 + 2\alpha_w\langle w_t - w^*_{\theta_t}, g_{\theta_t}(w_t, \mathcal{B}_t) - \bar{g}_{\theta_t}(w_t)\rangle \\
&\quad + 2\alpha_w^2\left\|g_{\theta_t}(w_t, \mathcal{B}_t) - \bar{g}_{\theta_t}(w_t)\right\|^2,
\end{aligned}$$

where (i) follows from the property $(w_t - w^*_{\theta_t})^T\bar{A}_t(w_t - w^*_{\theta_t}) \leq -\lambda\left\|w_t - w^*_{\theta_t}\right\|^2$ with some constant $\lambda > 0$ for any policy, which has been proved in Tsitsiklis and Van Roy (1997), Bhandari et al. (2018), Tu and Recht (2019), Xiong et al. (2020), and (ii) follows because $\|A\|^2 \leq 2(1 + \gamma^2)C_\phi^4 \leq 4C_\phi^4 := C_A^2$.

Taking the expectation over the actor and the critic parameters on both sides yields

$$\mathbb{E}\left\|w_{t+1} - w_{\theta_t}^*\right\|^2$$
$$\leq (1 - 2\alpha_w\lambda + 2\alpha_w^2 C_A^2)\mathbb{E}\left\|w_t - w_{\theta_t}^*\right\|^2 + 2\alpha_w\mathbb{E}\langle w_t - w_{\theta_t}^*, g_{\theta_t}(w_t, \mathcal{B}_t) - \bar{g}_{\theta_t}(w_t)\rangle$$
$$+ 2\alpha_w^2\mathbb{E}\left\|g_{\theta_t}(w_t, \mathcal{B}_t) - \bar{g}_{\theta_t}(w_t)\right\|^2$$
$$= (1 - 2\alpha_w\lambda + 2\alpha_w^2 C_A^2)\mathbb{E}\left\|w_t - w_{\theta_t}^*\right\|^2 + 2\alpha_w^2\mathbb{E}\left\|g_{\theta_t}(w_t, \mathcal{B}_t) - \bar{g}_{\theta_t}(w_t)\right\|^2. \tag{11}$$

Observe that

$$\mathbb{E}\left\|g_{\theta_t}(w_t, \mathcal{B}_t) - \bar{g}_{\theta_t}(w_t)\right\|^2$$
$$= \mathbb{E}\left\|\hat{A}_t w_t + \hat{b}_t - \bar{A}_t w_t - \bar{b}_t\right\|^2$$
$$\overset{(i)}{\leq} 3\mathbb{E}\left\|(\hat{A}_t - \bar{A}_t)(w_t - w_{\theta_t}^*)\right\|^2 + 3\mathbb{E}\left\|(\hat{A}_t - \bar{A}_t)w_{\theta_t}^*\right\|^2 + 3\mathbb{E}\left\|\hat{b}_t - \bar{b}_t\right\|^2$$
$$\leq 3\mathbb{E}\left\|\hat{A}_t - \bar{A}_t\right\|_F^2\left\|w_t - w_{\theta_t}^*\right\|^2 + 3\mathbb{E}\left\|\hat{A}_t - \bar{A}_t\right\|_F^2\left\|w_{\theta_t}^*\right\|^2 + 3\mathbb{E}\left\|\hat{b}_t - \bar{b}_t\right\|^2$$
$$\overset{(ii)}{\leq} \frac{12C_A^2}{M}\mathbb{E}\left\|w_t - w_{\theta_t}^*\right\|^2 + \frac{12(C_A^2\mathbb{E}\left\|w_{\theta_t}^*\right\|^2 + C_b^2)}{M}$$
$$\overset{(iii)}{\leq} \frac{12C_A^2}{M}\mathbb{E}\left\|w_t - w_{\theta_t}^*\right\|^2 + \frac{12(C_A^2 C_w^2 + C_b^2)}{M},$$

where (i) follows because $(x + y + z)^2 \leq 3x^2 + 3y^2 + 3z^2$, (ii) follows from Lemma 13 and $C_b := R_{\max}C_\phi \geq \|b\|$, and (iii) follows because $\left\|w_{\theta_t}^*\right\|^2 = \left\|\bar{A}_t^{-1}\bar{b}_t\right\|^2 \leq C_b/\lambda_A = R_{\max}C_\phi/\lambda_A := C_w$ by Assumption 6.

Substituting the above bound into (11), we have

$$\mathbb{E}\left\|w_{t+1} - w_{\theta_t}^*\right\|^2$$
$$\leq (1 - 2\alpha_w\lambda + 2\alpha_w^2 C_A^2)\mathbb{E}\left\|w_t - w_{\theta_t}^*\right\|^2 + 2\alpha_w^2\mathbb{E}\left\|g_{\theta_t}(w_t, \mathcal{B}_t) - \bar{g}_{\theta_t}(w_t)\right\|^2$$
$$\leq \left(1 - 2\alpha_w\lambda + 2\alpha_w^2 C_A^2 + \frac{24\alpha_w^2 C_A^2}{M}\right)\mathbb{E}\left\|w_t - w_{\theta_t}^*\right\|^2 + \frac{24\alpha_w^2(C_A^2 C_w^2 + C_b^2)}{M}.$$

Since $\alpha_w \leq \frac{\lambda}{2C_A^2}; M \geq \frac{48\alpha_w C_A^2}{\lambda}$, we further obtain

$$\mathbb{E}\left\|w_{t+1} - w_{\theta_t}^*\right\|^2$$
$$\leq \left(1 - 2\alpha_w\lambda + 2\alpha_w^2 C_A^2 + \frac{24\alpha_w^2 C_A^2}{M}\right)\mathbb{E}\left\|w_t - w_{\theta_t}^*\right\|^2 + \frac{24\alpha_w^2(C_A^2 C_w^2 + C_b^2)}{M}$$
$$\leq \left(1 - \frac{\alpha_w\lambda}{2}\right)\mathbb{E}\left\|w_t - w_{\theta_t}^*\right\|^2 + \frac{24\alpha_w^2(C_A^2 C_w^2 + C_b^2)}{M}. \tag{12}$$

Next, we use Young's inequality, and obtain

$$\mathbb{E}\left\|w_{t+1} - w_{\theta_{t+1}}^*\right\|^2$$
$$\leq \left(1 + \frac{1}{2(2/\lambda\alpha_w - 1)}\right)\mathbb{E}\left\|w_{t+1} - w_{\theta_t}^*\right\|^2 + (1 + 2(2/\lambda\alpha_w - 1))\mathbb{E}\left\|w_{\theta_t}^* - w_{\theta_{t+1}}^*\right\|^2$$
$$\overset{(i)}{\leq} \left(1 - \frac{\lambda\alpha_w}{4}\right)\mathbb{E}\left\|w_t - w_{\theta_t}^*\right\|^2 + \frac{4 - \lambda\alpha_w}{4 - 2\lambda\alpha_w} \cdot \frac{24\alpha_w^2(C_A^2 C_w^2 + C_b^2)}{M} + \frac{4}{\lambda\alpha_w}\mathbb{E}\left\|w_{\theta_t}^* - w_{\theta_{t+1}}^*\right\|^2$$
$$\overset{(ii)}{\leq} \left(1 - \frac{\lambda\alpha_w}{4}\right)\mathbb{E}\left\|w_t - w_{\theta_t}^*\right\|^2 + \frac{4 - \lambda\alpha_w}{4 - 2\lambda\alpha_w} \cdot \frac{24\alpha_w^2(C_A^2 C_w^2 + C_b^2)}{M} + \frac{12L_w^2}{\lambda\alpha_w}\mathbb{E}\left\|\theta_{t+1} - \theta_t\right\|^2 + \frac{24\kappa^2}{\lambda\alpha_w},$$
$$\tag{13}$$

where (i) follows from the bound derived in (12), and (ii) holds due to the fact that[5]

$$\mathbb{E}\left\|w_{\theta_t}^* - w_{\theta_{t+1}}^*\right\|^2$$

$$\leq 3\mathbb{E}\left\|w_{\theta_t}^* - w_{\xi_{\theta_t}}^*\right\|^2 + 3\mathbb{E}\left\|w_{\xi_{\theta_t}}^* - w_{\xi_{\theta_{t+1}}}^*\right\|^2 + 3\mathbb{E}\left\|w_{\xi_{\theta_{t+1}}}^* - w_{\theta_{t+1}}^*\right\|^2$$

$$\overset{(i)}{\leq} 3\kappa^2 + 3\mathbb{E}\left\|w_{\xi_{\theta_t}}^* - w_{\xi_{\theta_{t+1}}}^*\right\|^2 + 3\kappa^2$$

$$\overset{(ii)}{\leq} 6\kappa^2 + 3L_w^2\mathbb{E}\left\|\theta_{t+1} - \theta_t\right\|^2,$$

where (i) follows from the definition of $\kappa$ in (4), and (ii) follows from Lemma 14.

**Step II: Bounding cumulative tracking error via compatibility theorem for DPG.**

In this step, we bound the cumulative tracking error based on the dynamics of the tracking error from the last step. To this end, we need to first bound the difference between two consecutive actor parameters.

Observe that $\theta_{t+1} - \theta_t = \frac{\alpha_\theta}{M}\sum_{j=0}^{M-1}\nabla_\theta f_{\theta_t}(s'_{t,j}, \epsilon'_{t,j})\nabla_\theta f_{\theta_t}(s'_{t,j}, \epsilon'_{t,j})^T w_t = \alpha_\theta h_{\theta_t}(w_t, \mathcal{B}_t)$ and $\mathbb{E}\left\|h_{\theta_t}(w_t, \mathcal{B}_t)\right\|^2 \leq 2\mathbb{E}\left\|\nabla J(\theta_t)\right\|^2 + 2\mathbb{E}\left\|h_{\theta_t}(w_t, \mathcal{B}_t) - \nabla J(\theta_t)\right\|^2$. We proceed to bound (13) as follows

$$\mathbb{E}\left\|w_{t+1} - w_{\theta_{t+1}}^*\right\|^2$$

$$\leq \left(1 - \frac{\lambda\alpha_w}{4}\right)\mathbb{E}\left\|w_t - w_{\theta_t}^*\right\|^2 + \frac{4-\lambda\alpha_w}{4-2\lambda\alpha_w}\cdot\frac{24\alpha_w^2(C_A^2 C_w^2 + C_b^2)}{M} + \frac{12L_w^2}{\lambda\alpha_w}\mathbb{E}\left\|\theta_{t+1}-\theta_t\right\|^2 + \frac{24\kappa^2}{\lambda\alpha_w}$$

$$\leq \left(1 - \frac{\lambda\alpha_w}{4}\right)\mathbb{E}\left\|w_t - w_{\theta_t}^*\right\|^2 + \frac{48\alpha_w^2(C_A^2 C_w^2 + C_b^2)}{M} + \frac{24L_w^2\alpha_\theta^2}{\lambda\alpha_w}\mathbb{E}\left\|\nabla J(\theta_t)\right\|^2$$

$$\quad + \frac{24L_w^2\alpha_\theta^2}{\lambda\alpha_w}\mathbb{E}\left\|h_{\theta_t}(w_t, \mathcal{B}_t) - \nabla J(\theta_t)\right\|^2 + \frac{24\kappa^2}{\lambda\alpha_w}$$

$$\overset{(i)}{\leq}\left(1 - \frac{\lambda\alpha_w}{4} + \frac{72L_h^2 L_w^2\alpha_\theta^2}{\lambda\alpha_w}\right)\mathbb{E}\left\|w_t - w_{\theta_t}^*\right\|^2 + \frac{48\alpha_w^2(C_A^2 C_w^2 + C_b^2)}{M} + \frac{24L_w^2\alpha_\theta^2}{\lambda\alpha_w}\mathbb{E}\left\|\nabla J(\theta_t)\right\|^2$$

$$\quad + \frac{24L_w^2\alpha_\theta^2}{\lambda\alpha_w}\left(3L_h^2\kappa^2 + \frac{12L_f^4 C_{w_\xi}^2}{M}\right) + \frac{24\kappa^2}{\lambda\alpha_w}$$

$$\overset{(ii)}{\leq}\left(1 - \frac{\lambda\alpha_w}{8}\right)\mathbb{E}\left\|w_t - w_{\theta_t}^*\right\|^2 + \frac{24L_w^2\alpha_\theta^2}{\lambda\alpha_w}\mathbb{E}\left\|\nabla J(\theta_t)\right\|^2 + \frac{48\alpha_w^2(C_A^2 C_w^2 + C_b^2)}{M}$$

$$\quad + \frac{24L_w^2\alpha_\theta^2}{\lambda\alpha_w}\left(3L_h^2\kappa^2 + \frac{12L_f^4 C_{w_\xi}^2}{M}\right) + \frac{24\kappa^2}{\lambda\alpha_w}, \tag{14}$$

where (i) follows from Lemma 15, and (ii) follows because $\alpha_\theta \leq \frac{\lambda\alpha_w}{24L_h L_w}$.

We further take the summation over all iterations on both sides of (14) and have

$$\sum_{t=0}^{T-1}\mathbb{E}\left\|w_t - w_{\theta_t}^*\right\|^2$$

$$\leq \sum_{t=0}^{T-1}\left(1 - \frac{\lambda\alpha_w}{8}\right)^t\left\|w_0 - w_{\theta_0}^*\right\|^2 + \frac{24L_w^2\alpha_\theta^2}{\lambda\alpha_w}\sum_{t=0}^{T-1}\sum_{i=0}^{t-1}\left(1 - \frac{\lambda\alpha_w}{8}\right)^{t-1-i}\mathbb{E}\left\|\nabla J(\theta_t)\right\|^2$$

$$\quad + \left[\frac{48\alpha_w^2(C_A^2 C_w^2 + C_b^2)}{M} + \frac{24L_w^2\alpha_\theta^2}{\lambda\alpha_w}\left(3L_h^2\kappa^2 + \frac{12L_f^4 C_{w_\xi}^2}{M}\right) + \frac{24\kappa^2}{\lambda\alpha_w}\right]\sum_{t=0}^{T-1}\sum_{i=0}^{t-1}\left(1 - \frac{\lambda\alpha_w}{8}\right)^{t-1-i}$$

---

[5]In Xiong et al. (2022), they directly use $\mathbb{E}\left\|w_{\theta_t}^* - w_{\theta_{t+1}}^*\right\|^2 \leq L_w\mathbb{E}\left\|\theta_{t+1} - \theta_t\right\|^2$, which is not proven and is different from the inequality from Lemma 14: $\mathbb{E}\left\|w_{\xi_{\theta_t}}^* - w_{\xi_{\theta_{t+1}}}^*\right\|^2 \leq L_w\mathbb{E}\left\|\theta_{t+1} - \theta_t\right\|^2$. Here, we use the triangle inequality to give a bound for $\mathbb{E}\left\|w_{\theta_t}^* - w_{\theta_{t+1}}^*\right\|^2$.

$$\leq \frac{8\left\|w_0 - w_{\theta_0}^*\right\|^2}{\lambda\alpha_w} + \left[\frac{48\alpha_w^2(C_A^2 C_w^2 + C_b^2)}{M} + \frac{24L_w^2\alpha_\theta^2}{\lambda\alpha_w}\left(3L_h^2\kappa^2 + \frac{12L_f^4 C_{w_\xi}^2}{M}\right) + \frac{24\kappa^2}{\lambda\alpha_w}\right] \cdot \frac{8T}{\lambda\alpha_w}$$

$$+ \frac{192L_w^2\alpha_\theta^2}{\lambda^2\alpha_w^2}\sum_{t=0}^{T-1}\mathbb{E}\left\|\nabla J(\theta_t)\right\|^2. \tag{15}$$

**Step III: Overall convergence by canceling tracking error via actor's positive progress.**

In this step, we establish the overall convergence to a stationary policy by novel cancellation of the above cumulative tracking error via actor's update progress.

Based on Lemma 8, we have

$$\mathbb{E}[J(\theta_{t+1})] - \mathbb{E}[J(\theta_t)]$$

$$\geq \mathbb{E}\langle\nabla J(\theta_t), \theta_{t+1} - \theta_t\rangle - \frac{L_J}{2}\mathbb{E}\left\|\theta_{t+1} - \theta_t\right\|^2$$

$$= \alpha_\theta\mathbb{E}\langle\nabla J(\theta_t), h_{\theta_t}(w_t, \mathcal{B}_t)\rangle - \frac{L_J\alpha_\theta^2}{2}\mathbb{E}\left\|h_{\theta_t}(w_t, \mathcal{B}_t)\right\|^2$$

$$= \alpha_\theta\mathbb{E}\left\|\nabla J(\theta_t)\right\|^2 + \alpha_\theta\mathbb{E}\langle\nabla J(\theta_t), h_{\theta_t}(w_t, \mathcal{B}_t) - \nabla J(\theta_t)\rangle - \frac{L_J\alpha_\theta^2}{2}\mathbb{E}\left\|h_{\theta_t}(w_t, \mathcal{B}_t)\right\|^2$$

$$\overset{(i)}{\geq} \frac{\alpha_\theta}{2}\mathbb{E}\left\|\nabla J(\theta_t)\right\|^2 - \frac{\alpha_\theta}{2}\mathbb{E}\left\|h_{\theta_t}(w_t, \mathcal{B}_t) - \nabla J(\theta_t)\right\|^2$$

$$- \frac{L_J\alpha_\theta^2}{2}\mathbb{E}\left\|h_{\theta_t}(w_t, \mathcal{B}_t) - \nabla J(\theta_t) + \nabla J(\theta_t)\right\|^2$$

$$\geq \left(\frac{\alpha_\theta}{2} - L_J\alpha_\theta^2\right)\mathbb{E}\left\|\nabla J(\theta_t)\right\|^2 - \left(\frac{\alpha_\theta}{2} + L_J\alpha_\theta^2\right)\mathbb{E}\left\|h_{\theta_t}(w_t, \mathcal{B}_t) - \nabla J(\theta_t)\right\|^2$$

$$\overset{(ii)}{\geq} \left(\frac{\alpha_\theta}{2} - L_J\alpha_\theta^2\right)\mathbb{E}\left\|\nabla J(\theta_t)\right\|^2 - \left(\frac{\alpha_\theta}{2} + L_J\alpha_\theta^2\right)\left(3L_h^2\mathbb{E}\left\|w_t - w_{\theta_t}^*\right\|^2 + 3L_h^2\kappa^2 + \frac{12L_f^4 C_{w_\xi}^2}{M}\right)$$

$$\overset{(iii)}{\geq} \frac{\alpha_\theta}{4}\mathbb{E}\left\|\nabla J(\theta_t)\right\|^2 - \frac{3\alpha_\theta}{4}\left(3L_h^2\mathbb{E}\left\|w_t - w_{\theta_t}^*\right\|^2 + 3L_h^2\kappa^2 + \frac{12L_f^4 C_{w_\xi}^2}{M}\right), \tag{16}$$

where (i) follows because $x^T y \geq -\frac{1}{2}x^2 - \frac{1}{2}y^2$, (ii) follows from Lemma 15, and (iii) follows from the condition $\alpha_\theta \leq \frac{1}{4L_J}$.[6]

We next take the summation over all iterations on both sides of the above bound and obtain

$$\frac{\alpha_\theta}{4}\sum_{t=0}^{T-1}\mathbb{E}\left\|\nabla J(\theta_t)\right\|^2$$

$$\leq \mathbb{E}[J(\theta_{T+1})] - \mathbb{E}[J(\theta_0)] + \frac{3\alpha_\theta}{4}\left(3L_h^2\kappa^2 + \frac{12L_f^4 C_{w_\xi}^2}{M}\right)\cdot T + \frac{9\alpha_\theta L_h^2}{4}\sum_{t=0}^{T-1}\mathbb{E}\left\|w_t - w_{\theta_t}^*\right\|^2$$

$$\leq \frac{R_{\max}}{1-\gamma} + \frac{3\alpha_\theta}{4}\left(3L_h^2\kappa^2 + \frac{12L_f^4 C_{w_\xi}^2}{M}\right)\cdot T + \frac{9\alpha_\theta L_h^2}{4}\sum_{t=0}^{T-1}\mathbb{E}\left\|w_t - w_{\theta_t}^*\right\|^2. \tag{17}$$

Substituting the cumulative tracking error bound derived in (15) into (17) yields

$$\frac{\alpha_\theta}{8}\sum_{t=0}^{T-1}\mathbb{E}\left\|\nabla J(\theta_t)\right\|^2$$

$$\overset{(i)}{\leq} \left(\frac{\alpha_\theta}{4} - \frac{432L_h^2 L_w^2\alpha_\theta^3}{\lambda^2\alpha_w^2}\right)\sum_{t=0}^{T-1}\mathbb{E}\left\|\nabla J(\theta_t)\right\|^2$$

$$\leq \frac{R_{\max}}{1-\gamma} + \frac{3\alpha_\theta}{4}\left(3L_h^2\kappa^2 + \frac{12L_f^4 C_{w_\xi}^2}{M}\right)\cdot T + \frac{18\alpha_\theta L_h^2}{\lambda\alpha_w}\left\|w_0 - w_{\theta_0}^*\right\|^2$$

---

[6]Here, we highlight this condition on $\alpha_\theta$, which is missing from Theorem 1 of Xiong et al. (2022).

$$+ \left[ \frac{48\alpha_w^2(C_A^2 C_w^2 + C_b^2)}{M} + \frac{8L_w^2\alpha_\theta^2}{\lambda\alpha_w}\left(3L_h^2\kappa^2 + \frac{12L_f^4 C_{w_\xi}^2}{M}\right) + \frac{24\kappa^2}{\lambda\alpha_w} \right] \cdot \frac{18\alpha_\theta L_h^2 T}{\lambda\alpha_w},$$

where (i) follows from the condition $\alpha_\theta \leq \frac{\lambda\alpha_w}{24\sqrt{6}L_h L_w}$.

Finally, we have

$$\min_{t\in[T]}\mathbb{E}\left\|\nabla J(\theta_t)\right\|^2 \leq \frac{1}{T}\sum_{t=0}^{T-1}\mathbb{E}\left\|\nabla J(\theta_t)\right\|^2$$

$$\leq \left(\frac{8R_{\max}}{\alpha_\theta(1-\gamma)} + \frac{144L_h^2}{\lambda\alpha_w}\left\|w_0 - w_{\theta_0}^*\right\|^2\right)\cdot\frac{1}{T} + 6\left(3L_h^2\kappa^2 + \frac{12L_f^4 C_{w_\xi}^2}{M}\right)$$

$$+ \left[\frac{48\alpha_w^2(C_A^2 C_w^2 + C_b^2)}{M} + \frac{8L_w^2\alpha_\theta^2}{\lambda\alpha_w}\left(3L_h^2\kappa^2 + \frac{12L_f^4 C_{w_\xi}^2}{M}\right) + \frac{24\kappa^2}{\lambda\alpha_w}\right]\cdot\frac{144L_h^2}{\lambda\alpha_w}$$

$$= \frac{c_1}{T} + \frac{c_2}{M} + c_3\kappa^2,$$

where[7]

$$c_1 = \frac{8R_{\max}}{\alpha_\theta(1-\gamma)} + \frac{144L_h^2}{\lambda\alpha_w}\left\|w_0 - w_{\theta_0}^*\right\|^2, \tag{18}$$

$$c_2 = 72L_f^4 C_{w_\xi}^2 + \left[48\alpha_w^2(C_A^2 C_w^2 + C_b^2) + \frac{96L_w^2 L_f^4 C_{w_\xi}^2\alpha_\theta^2}{\lambda\alpha_w}\right]\cdot\frac{144L_h^2}{\lambda\alpha_w}, \tag{19}$$

$$c_3 = 18L_h^2 + \left[\frac{24L_w^2 L_h^2\alpha_\theta^2}{\lambda\alpha_w} + \frac{24}{\lambda\alpha_w}\right]\cdot\frac{144L_h^2}{\lambda\alpha_w}. \tag{20}$$

---

[7]In Xiong et al. (2022), $c_3 = 18L_h^2 + \frac{24L_w^2 L_h^2\alpha_\theta^2}{\lambda\alpha_w}$. Here, we fix this constant by adding the missing factor $\frac{144L_h^2}{\lambda\alpha_w}$ and the extra term $\frac{24}{\lambda\alpha_w}$ required in (13).

