# OpenReview forum: "Deep Policy Gradient Methods Without Batch Updates, Target Networks, or Replay Buffers"
_NeurIPS.cc/2024/Conference — NeurIPS 2024 poster_

### Official Review · Reviewer_2Tun · 2024-07-08

**Soundness:** 3
**Presentation:** 2
**Contribution:** 3
**Rating:** 6
**Confidence:** 3

**Summary:**

The authors consider resource-scare RL. In the considered setup, the agent is restricted to a small replay buffer, or single sample updates (incremental RL). The authors observe that traditional on-policy and off-policy algorithms (PPO, SAC, and TD3) fail to learn performing policies when the replay buffer size is restricted or in an incremental setup. The authors propose an incremental learning algorithm referred to as Action Value Gradient (AVG). AVG is implemented as SAC with a replay buffer of 1 with the addition of the following normalizations: feature normalization (pnorm as implemented in [1] and [2]), state normalization (as implemented in [3]), and return scaling (as implemented in [3]). The authors test their approach against incremental versions of PPO, SAC, and TD3 on 5 tasks from OpenAI Gym, 1 from DeepMind Control, 1 vision-based task referred to as "visual reacher" and 1 custom real-world task. The authors find that their method achieves significantly better performance than the baseline algorithms.

Although I like the paper very much, I believe that certain issues should be addressed in the published version of the manuscript. As such, I am willing to increase my score if the authors respond to the listed weaknesses and questions.

**Strengths:**

1. The authors tackle an important issue of RL in resource-scarce setups.
2. Big diversity of tasks (although with some questionable choices) with 30 seeds per task in the OpenAI Gym evaluation.
3. The problem is clear and well-motivated, and the paper is well-written.
4. Analysis of the effects of normalization on the performance of off-policy AC is timely and important.

**Weaknesses:**

1. I have a slight issue with how AVG is introduced. Namely, by dedicating some much space in Section 3. to orthogonal initialization, squashed normal policy, and maximum entropy formulation the authors imply that these are not standard in off-policy actor-critic (and they are used in modern SAC implementations e.g. [4]). These could be introduced in a single paragraph that would state the main differences between SAC and AVG.
2. Considering the above, would the authors agree that AVG is SAC without Clipped Double Q-learning and with added feature normalization, state normalization and return scaling? Such contribution would be fine for me since direct normalization of observations and return scaling is not standard in off-policy AC, and recent works are showing that applying various normalizations to SAC is promising (e.g. layer normalization [1]). If the authors agree with such sentiment, I believe that the manuscript (especially Section 3.) could be more upfront about the contributions of AVG. Furthermore, since the reparametrization policy gradient is not the main contribution of AVG, it might be more helpful for the reader if the name of the proposed technique was more related to its contribution.
3. I appreciate the amount of experiments that the authors report in their manuscript. However, I believe that the comparison against other normalization techniques that are commonly used in off-policy RL would be especially helpful for the reader. For example, it is not outlandish to hypothesize that the effects of state normalization and return scaling can be achieved by using layer normalization on the actor/critic modules (actually it was found to help exploding gradient norms that the authors report in Figure 4). Furthermore, some form of comparison of strategies for state and return scaling would help to better contextualize the strengths and weaknesses of the normalization scheme that AVG uses. Similarly, there are many initialization schemes besides orthogonal - if the authors want to claim contribution concerning using orthogonal initialization in off-policy AC then the authors should produce an argument for using it that is  the previous work
4. (nitpick) Mistake in Algorithm 1 there should be gradient descent for the critic and ascent for the actor (now it is ascent for both).

**Questions:**

1. I see that AVG does not use Clipped Double Q-learning, but does it use 2 critics still or a single one?
2. Is there a reason to consider a single task from the DMC benchmark?
3. How does the fact of being incremental affect the RP policy gradient? Does this on-policiness negatively affect the convergence properties (I would guess that e.g. SAC approximate convergence argument still works)?
4. Something unclear to me is whether AVG uses target networks. If not, this might also be an interesting point.

**Limitations:**

The authors discuss the limitations of their approach

---

> ### Author Rebuttal · Authors · 2024-08-07
>
> We are grateful to the reviewer for their time and commitment to evaluating our paper. We're glad the reviewer noted the importance of the problem setting, the diversity of tasks, and the use of 30 seeds per task. They also appreciated the clarity and writing quality. We're also pleased to read that the reviewer thinks our analysis of normalization effects on actor-critic methods is important and timely. Here, we first summarize the key points raised by the reviewer and respond to them.
>
> **Clarification of Contributions**
>
>  > Is AVG essentially SAC without Clipped Double Q-learning but with added feature normalization, state normalization, and return scaling?
>
> It is similar, but not the same. Here, we outline the components of both algorithms:
>
> | SAC | AVG |
> | - | - |
> | 1 actor | 1 actor |
> | 2 Q networks (i.e., double Q-learning) | 1 Q network |
> | 2 target Q networks | 0 target networks |
> | Learned $\alpha_{ent}$ | Fixed $\alpha_{ent}$ |
> | Replay buffer $\mathcal{B}$ | No buffers |
>
> In addition, SAC is off-policy, whereas AVG is on-policy. SAC samples an action and stores it in the buffer. Unlike AVG, SAC's action is not reused to update the actor.
>
> We appreciate the reviewer's constructive criticism about the clarity of the contributions. We will incorporate their feedback in our final manuscript.
>
> **Presentation of AVG**
>
> > Design choices such as orthogonal initialization, squashed normal policy, and maximum entropy formulation, which is standard in modern SAC implementations, can be condensed into a single paragraph
>
> We intended to outline every component utilized in our algorithm to enhance reproducibility. We agree with our reviewer that this information can be presented briefly.
>
> **Comparison with Other Normalization Techniques**
>
> > Can effects of state normalization and return scaling be achieved by using layer normalization on the actor/critic modules?
>
> As requested, we conducted additional ablations studying the effect of alternatives such as layer norm and RMS norm (Fig. 3 of PDF). In this comparison, our proposed techniques perform best when used with AVG.
>
> > Comparison of strategies for state and return scaling
>
> We chose Welford's online algorithm for normalizing observations since it is unbiased and maintains statistics over the entire dataset. In limited experiments, weighted methods biased toward more recent samples failed to work well in our experiments.
>
> Schaul et al. (2021) showcase the perils of clipping or normalizing rewards. Hence, we chose their simple approach, which scales the temporal difference error using a multiplicative scale factor.
>
> We agree with the reviewer that a different set of normalization techniques can potentially obtain similar, if not better, results. We only aim to highlight this issue and propose easy-to-use solutions.
>
> > Argument for using orthogonal weight initialization
>
> Our use of orthogonal weight initialization is consistent with popular implementations of SAC and TD3. However, empirical evidence is left for future work.
>
> **Typo in Algorithm 1**
>
> We thank the reviewer for catching this typo in our critic update rule in Algorithm 1. We will fix this in our final manuscript.
>
> ### Answers
>
> > Does AVG use two critics?
>
> AVG uses only one critic, which is an unclipped Q-network.
>
> > Is there a reason to consider a single task from the DMC benchmark?
>
> We consider DMC to determine the effectiveness of AVG in sparse reward environments. Additional results from environments such as Ball in Cup and Finger Spin will be included in the final manuscript.
>
> > How does the fact of being incremental affect the RP policy gradient? Does this on-policiness negatively affect the convergence properties?
>
> SAC's convergence argument results from Soft Policy Iteration (SPI) in the tabular setting. In this case, the critic converges to the true soft value function of the actor before the actor is updated. Under this regime, SPI converges to the optimal soft policy (Haarnoja et al., 2018). If we take the two-timescale perspective (Konda & Borkar, 1999; Konda & Tsitsiklis, 1999) on SAC, it can be seen as approximate (soft) policy iteration, which can serve as theoretical motivation for SAC. This argument would work for AVG as well.
>
> It is difficult to directly compare the theoretical properties of the actual implementations of AVG and SAC, given that they have many varying components. However, we may gain insights from analyzing algorithms with similar structures that only vary in the on-policiness/off-policiness.
>
> Following this direction, Xiong et al. (2022) show that for Deterministic Policy Gradient (DPG), up to the same system error that necessarily exists in actor-critic algorithms, on-policy DPG and off-policy DPG (with off-policy correction) have the same sample complexity of $\mathcal{O}(\epsilon^{-2})$ to achieve an $\epsilon$-accurate stationary policy. Note that existing implementations of deep off-policy algorithms do not include off-policy correction (unlike in Xiong et al. (2022)).
>
> We validated that these results can be extended to the case of reparametrized policy gradients, suggesting that changing from an off-policy distribution to an on-policy distribution for updating the critic would not induce a negative impact. In addition, reducing the batch size would impact sample complexity, but it is necessary for applications involving edge devices.
>
> Overall, a more careful theoretical understanding of AVG is essential and is left for future work.
>
> > Does AVG use target networks?
>
> AVG does not use target networks.
>
> We agree with the reviewer that it is indeed interesting that target Q-networks negatively impact incremental algorithms. We provide additional results in our PDF (Fig. 2) that show that SAC-1 can still fail when normalization and scaling tricks are incorporated.
>
> We believe we have addressed the reviewer's main concerns. If the reviewer agrees, we kindly request that they consider increasing their score accordingly.

---

> > ### Comment · Reviewer_2Tun · 2024-08-07
> > **Thank you for the rebuttal**
> >
> > Thank you for the clarifications. I am pleased with the additional experimental results, and I think that the results showing that normalization is crucial for incremental methods to work is very valuable. As such, I have increased my score of the manuscript.
> >
> > Having said this, I would kindly encourage the authors to implement the following changes if their paper is accepted:
> >
> > 1. Context wrt. to previous work - explicitly state that reparametrization trick, orthogonal initialization, and squashed normal policies are standard in off-policy actor-critic. It would be nice if the reader had the intuition that a good starting point for thinking about AVG is SAC with buffer size=1.
> >
> > 2. Appropriate visibility for the importance of normalization - preferably, the authors could expand the results presented in Figure 2 (perhaps more envs?) and put these results in a visible spot. Then, the authors should add something along the lines of "normalization helps these algorithms work in an incremental setting" to their contributions (although, this contribution should be worded very softly given that these results are limited). In my opinion, this observation might be interesting for the community.
> >
> > 3. Lack of target networks should be discussed more thoroughly - whereas not using target networks is not novel, the authors could expand the discussion of target networks in their setting. Ideally, authors could run a minimal experiment to test if there is something obvious that is wrong with using the target network in incremental setup (e.g. gradient norm explosion, estimation, etc)
> >
> > If the authors provide a tangible plan for implementing the above changes, I might consider revising the score again.

---

> > > ### Author Response · Authors · 2024-08-08
> > > **Response to Reviewer: Addressing Feedback and Outlining Planned Revisions**
> > >
> > > We thank the reviewer for carefully considering our response and increasing the score. We are also grateful for their prompt response and valuable feedback. We believe all three suggestions will increase the clarity and emphasize the contributions of our work. While a detailed plan will require careful forethought, we have outlined a brief plan for implementing the suggested changes:
> > >
> > > 1. In Section 3, it would make the exposition of the material clearer and more straightforward to state that the reparametrization trick, orthogonal initialization, and squashed normal policies are standard in off-policy actor-critic methods such as SAC and TD3. We also wish to highlight that we briefly motivated the use of squashed normal policies in Appendix A.5. In addition, we will further emphasize the similarities and differences between the learning update rules of AVG and SAC with a buffer size of 1 in the main text. To that extent, we will include the pseudocode for SAC-1 in the appendix for reference, using a similar notation to the main AVG algorithm.
> > >
> > > 2. We agree with the reviewer that the impact of normalization and scaling would be of wider interest to the community. Therefore, the paper would benefit by moving Figure 2 of our global-response PDF to Section 5 of the main paper. The revised version would contain two subsections discussing (a) the ablation study of normalization and scaling techniques used with AVG and (b) how normalization and scaling benefit other incremental algorithms. Note that the proposed subsection 5b would use softly worded descriptions to discuss these additional findings, only indicating the potential benefits of the proposed techniques. As suggested, we will include additional results from DM Control benchmarks.
> > >
> > > 3. *Discussion on the lack of target networks.* We plan to implement a variant of AVG that employs a target Q-network. Similar to SAC, we will use Polyak averaging to update the target Q-network: $\phi_{\text{target}} = (1 - \tau) \cdot \phi_{\text{target}} + \tau \cdot \phi$. We will run a minimal experiment varying $\tau$ between $(0, 1]$, where $\tau = 1$ implies that the target network is the same as the current Q-network at all timesteps. We will study the impact of a target Q-network by monitoring diagnostic metrics such as gradient norms on the Mujoco benchmarks previously used in our experiments.
> > >
> > > We look forward to hearing the reviewer's thoughts on our plan for the next steps. If the plan seems reasonable, we kindly request the reviewer to consider increasing their score as they suggested.

---

> > > > ### Comment · Reviewer_2Tun · 2024-08-11
> > > > **Thank you for the additional information**
> > > >
> > > > Thank you for outlining the plan for changes in the camera-ready version. I believe that the paper deserves to be published and as such I am happy to increase my score to 6 - congratulations!

---

> > > > > ### Author Response · Authors · 2024-08-12
> > > > > **Thank you for the response**
> > > > >
> > > > > Thank you very much for your constructive feedback and for increasing the score of our paper. We are pleased to hear that you believe our work deserves to be published!

---

### Official Review · Reviewer_gtgH · 2024-07-13

**Soundness:** 3
**Presentation:** 4
**Contribution:** 3
**Rating:** 7
**Confidence:** 4

**Summary:**

The paper address the problem of incremental reinforcement learning in low compute regimes. It introduces an algorithm Action Value Gradient (AVG), that uses the entropy regularized actor-critic learning method for solving the task in an incremental fashion. The $\textbf{re-parameterization}$ (RP) trick is used to estimate the gradients (for lower variance) and the actor/critic are updated every time-step. Furthermore, many normalization techniques are used to stabilize the learning process. The experiments are carried out on a variety of tasks in simulation. In addition, experiments are carried out on two real robot settings. The evaluation shows the effectiveness of AGV in comparison to an incremental learning method (IAC) and other off-the-shelf RL algorithms modified to do incremental learning. An ablation study is also provided to show the effectiveness of the normalization on the gradients of the incremental update.

**Strengths:**

The paper is well-written. The problem the paper addresses is important to the subcommunities of continual learning and adaptive learning. In addition, the reviewer thinks this problem is also important for search-and-rescue scenarios, ocean/sea exploration, etc.

The paper combines very well the existing methods to solve a new problem:

- Using the RP trick for reduced variance for incremental learning is novel with standard RL tools.
- Combining various normalization strategies to bound the inputs, actions, returns and gradients for learning stability.

The authors have rigorously experimented in simulation and in real settings.

1. The proposed method (AVG) beats the incremental learning baseline IAC on almost all of the tasks. An experiment is performed for incremental RL from only visual inputs, where AVG outperforms IAC.
2. The real robot trials further validate the method and is an interesting result (albeit simple tasks and a constrained task space) for the community as learning with few trials on the real robot setting can imply adapting on the fly.

The authors provide extensive implementation detail to replicate the results and also provide code for the same.

**Weaknesses:**

The paper shows interesting results, however the reviewer thinks there are certain weaknesses in the experiments:

1. The comparisons with SAC-1, SAC-100, TD3-1 may not be fair because these may not be using similar normalizing tricks to stabilize learning. Since these algorithms were developed for large batch training and large replay buffers, the reviewer does not see this as a fair comparison.
2. Comparing with other incremental learning algorithms on edge devices (or low compute) by reducing the buffer and batch sizes. The authors mention that these existing methods use cloud to store replay buffers and train. However, these comparisons would be more aligned with the proposed method.
3. The paper is restricted to learning from one new data-point every step with no buffer size. An ablation study how much low compute device could be exploited would make the paper stronger.  For example, a study of incremental learning where having a buffer size n, batch-size m and varying sizes while still being able to work on the low compute device would provide more insights on the incremental learning problem in such devices.

**Questions:**

As listed in the weakness section, The reviewer suggests changes in this order of priority:

1. Comparison with other incremental/continual learning methods (mentioned in the paper) in the low compute regime (The authors may have to again use the same normalization tricks).
2. Making the comparison with SAC-1, SAC-100, TD3-1 with the same normalizations.
3. An ablation with relatively low batch and buffer sizes.

This would help re-evaluate the review more towards the positive.

**Limitations:**

Yes, the authors talk about some of the limitations that the reviewer is in agreement with.
The work addresses learning on low compute devices that suggests a positive societal impact in terms resources spent on large compute.

---

> ### Author Rebuttal · Authors · 2024-08-07
>
> We are grateful to the reviewer for their time and commitment to evaluating our paper. We are pleased to read that the reviewer thinks our paper is well-written and that the problem we focus on is important to the continual learning and adaptive learning communities as well as for applications in search-and-rescue scenarios and ocean/sea exploration. Here, we jointly address the reviewer's points and questions in the order of priority suggested by the reviewer.
>
>
> ### Answers
>
> > Comparison with other incremental/continual learning methods (mentioned in the paper) in the low compute regime (The authors may have to again use the same normalization tricks).
>
> We have results incorporating normalization and scaling with incremental actor critic (IAC) listed in Appendix B.2. For further evidence, we also provide additional results that compare AVG and IAC with normalization and scaling in our global response PDF (see Fig. 2). We demonstrate that IAC also benefits from normalization and scaling.
>
> > Making the comparison with SAC-1, SAC-100, TD3-1 with the same normalizations.
>
> The reviewer raised concerns that the comparisons with SAC-1, SAC-100, and TD3-1 may be unfair since they do not use similar normalization and scaling techniques to stabilize learning. As requested, we have additional results incorporating normalization and scaling with SAC-1, SAC-100 and TD3-1 and comparing them against AVG. Please see our results in the global response PDF (Fig. 2). Notably, SAC-1 combined with normalization and scaling performs effectively on two tasks but fails or even diverges on the others.
>
> We would like to emphasize that the primary purpose of these experiments (Fig. 2 in our paper) is to show that existing methods cannot be naively used off-the-shelf in the incremental learning setting. Currently, there is no clear evidence in the literature, and as such, the community remains unaware that these batch methods fail catastrophically in the incremental learning setting. Therefore, our experiment was justified, given the research question we were addressing.
>
> > An ablation with relatively low batch and buffer sizes.
>
> This is an interesting question where the reviewer asks how best to utilize a resource-constrained device. This study needs to be carefully designed and conducted appropriately. We agree with the initial direction provided by the reviewer as well:
>
> *“An ablation study how much low compute device could be exploited would make the paper stronger. For example, a study of incremental learning where having a buffer size n, batch-size m and varying sizes while still being able to work on the low compute device would provide more insights on the incremental learning problem in such devices.”*
>
> However, given the time constraints, this is out of the scope of our paper. That said, we definitely believe this should be pursued in future work!
>
> We believe we have addressed the reviewer's main concerns. If the reviewer agrees, we kindly request that they consider increasing their score accordingly.

---

> > ### Author Response · Authors · 2024-08-12
> >
> > We thank the reviewer once again for their insightful comments and questions. We kindly ask if our response has addressed their main concerns and, if so, whether they would consider endorsing the paper for higher exposure by increasing the score.

---

> > ### Comment · Reviewer_gtgH · 2024-08-12
> >
> > Thanks for the response!
> >
> > The authors address the concerns the reviewer raised. In addition, the experiments on different algorithms (IAC+, SAC-1+, TD3-1+) in the continual learning setting show the effectiveness and generality of the method. This work is incremental and might see an adoption for future work in the area. The reviewer will change the rating to 7 for the reason above.

---

> > > ### Author Response · Authors · 2024-08-13
> > >
> > > Thank you very much for your constructive feedback and for increasing the score of our paper.

---

### Official Review · Reviewer_8wnP · 2024-07-14

**Soundness:** 3
**Presentation:** 3
**Contribution:** 3
**Rating:** 6
**Confidence:** 3

**Summary:**

In this paper, the authors propose a deep RL algorithm named AVG (Average Value Gradient) which uses incremental learning. This allows eliminating storing experiences in replay buffer for training models. They also combine tricks like penultimate normalization and return scaling for robustness. AVG also does not need any target Q-model for updating, thus saving space even further. The experiments on Mujoco show improved performance of AVG over other deep RL algorithms such as SAC, TD3 etc. The authors also show experiments on resource-constrained robotic tasks.

**Strengths:**

The proposed algorithm does not need replay buffer to store experiences. Instead, it uses incremental learning with deep neural networks to represent actor and critic. Combined with techniques like reparameterization, scaling and normalizations improves the robustness of the algorithm.
The proposed algorithm shows good performance on deep RL benchmarks.

**Weaknesses:**

The algorithm is sensitive to hyper-parameters, network size etc.
The algorithm also has poor sample efficiency as pointed out by the authors.

**Questions:**

My main concern is that deep RL algorithms have been shown to be very sensitive to hyper-parameters, network size, reward scaling and even random seeds! And now that we have AVG with incremental update, this might exacerbate this issue. Is there a trade-off between not storing experiences/target models and robustness?

**Limitations:**

The authors have adequately addressed the limitations. I have raised my concerns above.

---

> ### Author Rebuttal · Authors · 2024-08-07
>
> We are grateful to the reviewer for their time and commitment to evaluating our paper. We are pleased that the reviewer recognizes the value of eliminating replay buffers and target Q-networks to develop simpler, more computationally efficient learning algorithms. Below, we address the points raised by the reviewer.
>
> **Sensitivity to Hyperparameters**
>
> We agree with the reviewer that deep RL algorithms are notoriously sensitive to hyperparameters, including the random seed. One way to mitigate hyperparameter tuning is to identify a single set of hyperparameters that is robust across a wide range of tasks. While this set may not be optimal for each task individually, it may perform well across many tasks. Popular algorithms like DreamerV3, TD-MPC2, SAC, and PPO often showcase their robustness by using a single hyperparameter configuration that works well across multiple benchmark environments.
>
> Out of the 300 configurations presented in the paper, one consistently ranked in the top 10 across four environments and achieved effective performance, coming within 80\% of the best performance in each. We conducted additional experiments with this configuration, running 30 different seeds for 10 million steps. The learning curves for these experiments are shown in Figure 1 of the global response PDF.
> The hyper-parameter configuration is provided below:
>
> | Parameter                      | Value   |
> |--------------------------------|---------|
> | $\alpha_{actor}$               | 0.0063  |
> | $\alpha_{Q}$                   | 0.0087  |
> | Betas for Adam Optimizer       | [0, 0.999] |
> | Entropy coefficient $\alpha_{ent}$ | 0.07   |
> | Discount factor $\gamma$ | 0.99 |
> | Num. hidden units              | 256     |
> | Num. layers                    | 2       |
>
> We hope this addresses the reviewer's concerns.
>
> **Sample Efficiency**
>
> We acknowledge that there is potential to improve AVG's sample efficiency. Off-policy methods like Soft Actor-Critic (SAC) indeed achieve higher sample efficiency by utilizing replay buffers, which allow for multiple uses of each sample. However, these methods are often not suitable for deployment on microcontrollers and resource-limited systems due to their memory requirements and computational complexity.
>
> Our current focus is on fixing incremental deep policy gradient methods, which we consider a crucial first step. Improving the sample efficiency of these incremental methods remains an important area for future research.
>
> ### Answers
>
> > My main concern is that deep RL algorithms have been shown to be very sensitive to hyperparameters, network size, reward scaling, and even random seeds! And now that we have AVG with incremental updates, this might exacerbate this issue.
>
> We would like to emphasize that we did not perform any hyper-parameter tuning for the real robot experiments. Due to the significant cost and time requirements associated with real robot experiments, it is impractical to conduct hyper-parameter tuning directly on real robots. Instead, we selected an effective hyper-parameter configuration based on our simulation experiments and applied it to both UR-Reacher-2 and Create-Mover tasks (see Fig. 8 of paper). Additionally, we have previously mentioned that we identified a set of hyper-parameters that perform well across various tasks, further supporting the robustness of our chosen configurations.
>
> *Effect of network size*
>
> It is important to clarify that our current work does not demonstrate or claim any sensitivity of our algorithm to network size. All our experiments involve neural networks with two hidden layers, 256 units each. Investigating the impact of network size on our algorithm's performance and efficiency remains an open question for future work.
>
> > Is there a trade-off between not storing experiences/target models and robustness?
>
> A large replay buffer can place a heavy memory burden, especially for onboard and edge devices with limited memory. Therefore, we need computationally efficient alternatives to replay buffers that can help consolidate learned experiences over time. So, we agree that there is a trade-off here. Lan et al. (2022) explore this trade-off by introducing memory-efficient reinforcement learning algorithms based on the deep Q-network (DQN) algorithm. Their approach can reduce forgetting and maintain high sample efficiency by consolidating knowledge from the target Q-network to the current Q-network while only using small replay buffers. We recognize that the question is intriguing and remains an open area for future research.
>
> We believe we have addressed the reviewer's main concerns. If the reviewer agrees, we kindly request that they consider increasing their score accordingly.
>
> **References**
> - Lan, Q., Pan, Y., Luo, J., & Mahmood, A. R. (2022). Memory-efficient reinforcement learning with value-based knowledge consolidation. arXiv preprint arXiv:2205.10868.

---

> > ### Author Response · Authors · 2024-08-12
> >
> > We thank the reviewer once again for their insightful comments and questions. We kindly ask if our response has addressed their main concerns and, if so, whether they would consider endorsing the paper for higher exposure by increasing the score.

---

> > ### Comment · Reviewer_8wnP · 2024-08-12
> >
> > Thanks for the details. I am pleased that for the hyper-parameters selection, there was no "over-optimization" being done. My questions have been answered and I am willing to raise my score.

---

> > > ### Author Response · Authors · 2024-08-13
> > >
> > > Thank you for your response and for raising the score of our paper.

---

### Official Review · Reviewer_bZo8 · 2024-07-21

**Soundness:** 3
**Presentation:** 4
**Contribution:** 3
**Rating:** 6
**Confidence:** 4

**Summary:**

This work presents a well-grounded algorithm advancement in incremental deep policy gradient learning, which builts on Action Value Gradient Theorem. Extensive simulated and real-world experiments and ablations demonstrate the superiority of the proposed AVG algorithm.

**Strengths:**

1. This work is a novel method of deep policy gradient for incremental learning without replay buffer or batch updates.
2. The work tackles an important challenge in applications of real-world robot learning.
3. The colab notebook can be helpful for readers to understand and reproduce the results.

**Weaknesses:**

The motivation of AVG instead of IAC method is limited to some extent. Inclusion of comparisons of LR and RP (IAC, AVG) methods could be beneficial. It seems unclear that, as evidenced by Figure 3, whether the normalization & scaling techniques are more essential for incremental deep policy gradient learning, since that IAC without these techniques shows reasonable performance.

**Questions:**

1. Can the authors provide comparison and detailed analysis on LR and RP? In Fig.14 at Sec B.2, normalization & scaling techniques on IAC also indicate great performance for LR-based method.
2. How about normalization & scaling techniques applying to SAC-1?
3. Take into consideration the application scenario, instead of learning from scratch, it might be beneficial to further discuss on the performance of different incremental learning methods with pre-trained policies.
4. Can the authors add explanations of the relation between the proposed method and real-world reinforcement learning methods, e.g. [1]?

[1] Smith, L., Kostrikov, I., & Levine, S. (2022). A walk in the park: Learning to walk in 20 minutes with model-free reinforcement learning. arXiv preprint arXiv:2208.07860.

**Limitations:**

1. It world be great to refine Fig.8 & Fig.18's legend and analysis (e.g. in Fig.18, the discussion of AVG with full PPO & SAC)
2. Missing references of some supplementary sections in main paper.

Minor. Typo in Fig.1's text of y-axis.

---

> ### Author Rebuttal · Authors · 2024-08-07
>
> We are grateful to the reviewer for their time and commitment to evaluating our paper. We are pleased that the reviewer thinks our proposed method, AVG, is a novel, "well-grounded algorithm advancement" that tackles an important challenge in real-world applications. We are also happy that the reviewer appreciates the extensive set of experiments both in simulation and on real-world robots. Here, we first summarize the key points/concerns raised by the reviewer and respond to them.
>
> **The Motivation for Choosing AVG over IAC**
>
> Our motivation for introducing AVG is two-fold:
> - Reparameterization gradients offer an alternative approach to estimating the gradient and have been observed to demonstrate lower variance in practice (Greensmith et al. 2004, Fan et al. 2015, Lan et al. 2021).
> - There is a gap in the existing literature, as no incremental RP gradient method currently exists.
>
> We mention this in the introduction, but we can state this more clearly in the final manuscript.
>
>
> **The Essentiality of Normalization and Scaling**
>
> > From Figure 3, it is unclear whether normalization and scaling techniques are essential for incremental deep policy gradient learning, given that IAC without these techniques shows reasonable performance.
>
> We believe there is some misunderstanding here. IAC performs very poorly in Figure 3, exhibiting high variance and low mean performance. The vision-based reacher task used here is an adaptation of the Deepmind Reacher task with a reward of $-1$ every step until the arm's end-effector reaches the goal. The return indicates that IAC has failed to learn a good policy that can successfully reach the target. We will further clarify this point in the final manuscript.
>
> Normalization and scaling are not necessary for simple environments like Reacher-v4 and Acrobot-v4. However, they are crucial for robust performance on complex simulation benchmarks and real-world robot tasks.
>
>
> ### Answers
> > Can the authors provide comparison and detailed analysis on LR and RP? In Fig.14 at Sec B.2, normalization & scaling techniques on IAC also indicate great performance for LR-based method.
>
> Yes. Normalization and scaling do benefit IAC as well. We have included additional experimental evaluation that compares AVG and IAC, both benefiting from normalization and scaling (Fig. 2 in global response PDF).
>
>
> > How about normalization & scaling techniques applying to SAC-1?
>
> Since there are multiple implementations of SAC with different design choices, we first briefly list the important components used in our implementation of SAC:
> - One actor network.
> - Two Q networks (i.e., double Q-learning).
> - Two target Q networks.
> - Squashed normal policy.
> - Automatically tuned entropy coefficient.
> - Replay buffer.
>
>
> With SAC-1, we only get rid of a replay buffer, using only the most recent sample of updating all the networks. SAC-1, combined with normalization and scaling, performs effectively on two tasks but fails or even diverges on the others (Fig. 2 in global response PDF). Target networks may hinder incremental learning. Removing these will bring the algorithm closer to AVG.
>
>
> > Take into consideration the application scenario, instead of learning from scratch, it might be beneficial to further discuss on the performance of different incremental learning methods with pre-trained policies.
>
> In this work, we priortized learning from scratch to keep the study focused. We aim to explore this interesting direction in future work.
>
> >Can the authors add explanations of the relation between the proposed method and real-world reinforcement learning methods, e.g. [1]?
>
> This is an interesting question. Smith et al. (2022) demonstrated real-time learning capabilities on a quadruped locomotion task. They use a batch method with a replay buffer that combines DroQ [2] and layer normalization [3] with SAC. They utilize a laptop with NVIDIA GeForce RTX 2070 GPU for onboard inference and learning. There are some notable differences compared to our  setup:
> - Their learning was not onboard as the laptop was too bulky to be carried on the robot. In their videos, it can be clearly seen that the robot is tethered to a laptop using a wired connection. In contrast, our mobile robot experiments involve a small edge device directly attached to the iRobot Create2.
> - Our edge device, Jetson Nano, is significantly less powerful than their laptop. Their approach involves highly optimized JAX implementations and other code-level optimizations that reduce the computational time of an otherwise intensive batch gradient method. In contrast, our method, implemented in PyTorch, is incremental and computationally cheap by design, without many code-level optimizations.
> - [1] also constrains the joints to limit the robot's exploration, enabling quick learning. In contrast, we do not apply this constraint and demonstrate that AVG can perform well without extensive engineering.
>
> We acknowledge that there can be trade-offs between batch and incremental methods depending on memory availability and compute resources onboard a robot. With highly optimized software and hardware, running constrained variants of batch methods might be possible. We only suggest that our incremental methods would be much more amenable to real-time learning compared to current batch methods.
>
> We believe we have addressed the reviewer's main concerns. If the reviewer agrees, we kindly request that they consider increasing their score accordingly.
>
> *References*
> 1. Smith, L., Kostrikov, I., & Levine, S. (2022). A walk in the park: Learning to walk in 20 minutes with model-free reinforcement learning. arXiv preprint arXiv:2208.07860.
> 2. T. Hiraoka, T. Imagawa, T. Hashimoto, T. Onishi, and Y. Tsuruoka, "Dropout q-functions for doubly efficient reinforcement learning," International Conference on Learning Representations (ICLR), 2022.
> 3. Ba, J. L., Kiros, J. R., & Hinton, G. E. (2016). Layer normalization. arXiv preprint arXiv:1607.06450.

---

> > ### Author Response · Authors · 2024-08-12
> >
> > We thank the reviewer once again for their insightful comments and questions. We kindly ask if our response has addressed their main concerns and, if so, whether they would consider endorsing the paper for higher exposure by increasing the score.

---

> ### Comment · Reviewer_bZo8 · 2024-08-13
> **Thank you for the rebuttal!**
>
> Thank you for the effort, and I am pleased with the added explanation and experimental comparisons. I also agree with the authors that incremental methods would be much more amenable to real-time learning compared to current batch methods. My concern left is
>
> > Yes. Normalization and scaling do benefit IAC as well. We have included additional experimental evaluation that compares AVG and IAC, both benefiting from normalization and scaling (Fig. 2 in global response PDF).
>
> > Normalization and scaling are not necessary for simple environments like Reacher-v4 and Acrobot-v4. However, they are crucial for robust performance on complex simulation benchmarks and real-world robot tasks.
>
> Since IAC+ "IAC with normalization & scaling" do benefit from normalization & scaling, would it be beneficial to conduct all experiments with IAC+ (or AVG-) as one baseline on complex simulation benchmarks and real-world robot tasks  (e.g. Fig.3's)? ( To make the contribution of AVG vs. normalization & scaling techniques more clear.)
>
> I lean towards keeping the current score of weak accept given the above major experimental consideration.

---

> > ### Author Response · Authors · 2024-08-13
> >
> > > Since IAC+ "IAC with normalization & scaling" do benefit from normalization & scaling, would it be beneficial to conduct all experiments with IAC+ (or AVG-) as one baseline on complex simulation benchmarks and real-world robot tasks (e.g. Fig.3's)?
> >
> > We wish to highlight that AVG- is already discussed in our ablation study concerning the impact of normalisation and scaling on AVG. In previous rebuttal responses to Reviewer 2, we mentioned that we will add additional results from DM Control Suite where we compare AVG with other incremental methods including IAC+ and SAC-1+. As suggested, we will also evaluate IAC+ on both the pixel-based task and the real robot reacher task involving the UR5 robot arm. We will provide these additional results in the final manuscript.
> >
> > We hope this addresses the concerns raised by reviewer.

---

### Author Rebuttal · Authors · 2024-08-07

We thank our reviewers for their insightful comments and questions.

We are pleased to see that reviewers recognized several notable strengths of our paper: 1) the novelty and well-grounded approach of AVG, 2) its relevance to continual and adaptive learning, with potential applications in search-and-rescue and ocean exploration, 3) the extensive experiments conducted in both simulations and real-world robotic scenarios, 4) the elimination of replay buffers and target Q-networks, resulting in simpler algorithms, 5) the timely analysis of normalization effects on actor-critic methods, and 6) the clarity and quality of our writing.

We note that the reviewers had raised some questions, primarily regarding the impact of normalization and scaling on other incremental methods like IAC, SAC-1, and TD3-1, as well as the clarity of our contributions and AVG's sensitivity to hyperparameters. We have addressed them thoroughly and provided additional results in the PDF. We also thank the reviewers for suggesting the application of normalization and scaling techniques to other incremental algorithms, which strengthens our paper and further supports its claims.

We look forward to your responses and are happy to answer any additional questions.


Sincerely,

The Authors

---

### Decision · Program_Chairs · 2024-09-25

**Decision:**

Accept (poster)

**Comment:**

The paper proposes a novel approach to incremental learning in reinforcement learning environments (e.g. simulated robot control tasks). The approach, Action Value Gradient, along with normalization and scaling schemes, is shown to be practical and effective on multiple robots.

The reviewers agreed that the problem is well-motivated, the solution is novel and technically sound, and the experiments are convincing. The reviewers asked about the impact of normalization and scaling on other incremental algorithms, which the authors demonstrated thoroughly in their response. Including the additional results and discussion from the reviewer-author discussion will substantially strengthen the paper.